# Watching a double strand break repair polymerase insert a pro-mutagenic oxidized nucleotide

Joonas A. Jamsen [1], Akira Sassa[2], David D. Shock[1], William A. Beard[1] & Samuel H. Wilson [1]✉

Oxidized dGTP (8-oxo-7,8-dihydro-2′-deoxyguanosine triphosphate, 8-oxodGTP) insertion by DNA polymerases strongly promotes cancer and human disease. How DNA polymerases discriminate against oxidized and undamaged nucleotides, especially in error-prone double strand break (DSB) repair, is poorly understood. High-resolution time-lapse X-ray crystal-lography snapshots of DSB repair polymerase μ undergoing DNA synthesis reveal that a third active site metal promotes insertion of oxidized and undamaged dGTP in the canonical *anti*-conformation opposite template cytosine. The product metal bridged O8 with product oxy-gens, and was not observed in the *syn*-conformation opposite template adenine ($A_t$). Rotation of $A_t$ into the *syn*-conformation enabled undamaged dGTP misinsertion. Exploiting metal and substrate dynamics in a rigid active site allows 8-oxodGTP to circumvent polymerase fidelity safeguards to promote pro-mutagenic double strand break repair.

[1] Genome Integrity and Structural Biology Laboratory, National Institute of Environmental Health Sciences, National Institutes of Health, Research Triangle Park, NC 27709, USA. [2] Laboratory of Chromatin Metabolism and Epigenetics, Graduate School of Science, Chiba University, Chiba, Japan. ✉email: wilson5@niehs.nih.gov

Reactive oxygen species (ROS) are generated by numerous endogenous and exogenous processes[1]. Accumulation of excess cellular ROS during oxidative stress damages free nucleotide (dNTP) pools[2]. 8-oxo-7,8-dihydro-2′-deoxyguanosine triphosphate (8-oxodGTP), a common oxidized nucleotide, accounts for up to ~10% of total dGTP in oxidative stress conditions[3]. The 8-oxodG base readily adopts the *syn*-conformation allowing hydrogen bond formation with template adenine ($A_t$) using the Hoogsteen edge (Fig. 1a). Watson–Crick hydrogen bonding opposite template cytosine ($C_t$) is enabled by the *anti*-conformation. Most DNA polymerases (pols) discriminate only weakly against 8-oxodGTP[4,5], leading to its widespread promutagenic insertion into the genome. Despite degradation by pyrophosphorylase hMTH1[6–8], elevated genomic 8-oxodG levels drive cancer, aging, and human disease[9]. The deleterious effects of 8-oxodG are highlighted by the numerous cellular defense mechanisms evolved to attempt to suppress its accumulation[10].

Structural characterization of 8-oxodGTP insertion has only been reported for X-family pols β[5,11–14] and λ[15], and insertion opposite $C_t$ is limited to pol β[12]. Insertion opposite $A_t$ in *syn*-conformation is more efficient than insertion opposite $C_t$ for pols β[12] and λ[15]. The pol β pre-chemistry ternary complex was destabilized by a clash between $P_\alpha$ and O8 in the *anti*-conformation[12]. A third or product metal in the product complex opposite either template base was observed, but its role was unclear. The product metal has been suggested to either prevent the reverse reaction[16–18], or be essential for the forward reaction[19]. DNA polymerase μ incorporates nucleotides in the non-homologous end-joining (NHEJ) pathway of double-strand break (DSB) repair[20]. Error-prone synthesis by pol μ often introduces mismatches, as well as short deletions or insertions, at repaired sites. Pol μ is expected to frequently encounter oxidized nucleotides in its biological roles. The mechanism of low-fidelity synthesis by pol μ, however, is poorly understood. We employed time-lapse X-ray crystallography to obtain snapshots of 8-oxodGTP insertion by pol μ on a model DSB substrate. The results uncover unique molecular strategies for modulation of oxidized and undamaged dGTP insertion by this X-family DSB repair polymerase.

## Results

**Insertion infidelity**. DNA polymerases preferentially insert 8-oxodGTP opposite a templating adenine ($A_t$) base, instead of cytosine ($C_t$) (Fig. 1a)[5,11]. To understand discrimination by pol μ, we determined catalytic efficiencies for dGTP and 8-oxodGTP insertion opposite both $C_t$ and $A_t$, with $Mg^{2+}$ or $Mn^{2+}$ as metal ion co-factor. $Mg^{2+}$ is commonly employed as the metal ion cofactor with DNA polymerases, while $Mn^{2+}$ has been suggested to be the physiological metal cofactor of pol μ[21–23]. As shown in Fig. 1b (Supplementary Fig. 1 and Supplementary Table 1), 8-oxodGTP was preferentially inserted opposite $A_t$ (~25-fold) in the presence of $Mg^{2+}$. This specificity arose from an increase in efficiency of 8-oxodGTP insertion opposite $A_t$, and decreased efficiency opposite $C_t$, compared to dGTP insertion. The efficiency of $Mn^{2+}$-mediated 8-oxodGTP insertion opposite $A_t$ was hardly affected compared to $Mg^{2+}$, whereas insertion opposite $C_t$ was increased ~120-fold. $Mn^{2+}$ therefore dramatically altered discrimination and promoted insertion opposite $C_t$. Substituting $Mn^{2+}$ for $Mg^{2+}$ resulted in increased catalytic efficiency and reduced discrimination for dGTP insertion opposite both templates.

**dGTP insertion opposite cytosine and adenine**. To gain insight into the influence of C8 oxidation on the *anti/syn*-equilibrium and on discrimination of nucleotide insertion, we first characterized undamaged dGTP insertion. We grew crystals of pol μ in complex with a model DSB substrate containing a single nucleotide gap with a templating C ($C_t$)[24,25]. We soaked pol μ-DNA binary complex crystals in a cryo solution containing 1 mM dGTP in the presence of catalytically inert $Ca^{2+}$ (10 mM) to obtain pol μ-DNA-dGTP ternary complex crystals. We then solved the structure of the $Ca^{2+}$-bound pre-catalytic ground state (GS) ternary complex of dGTP opposite template C ($dGTP:C_t$) (Fig. 2a and Supplementary Table 2). The guanine base of dGTP (*anti*) forms Watson–Crick hydrogen bonds with $C_t$ in *anti*-conformation (Fig. 2a, b). The methylene backbone of Lys438 stabilizes the guanine base. Arg445 forms hydrogen bonds with O2 of $C_t$(*anti*) (Fig. 2b). The nucleotide ($Ca_n$) and catalytic

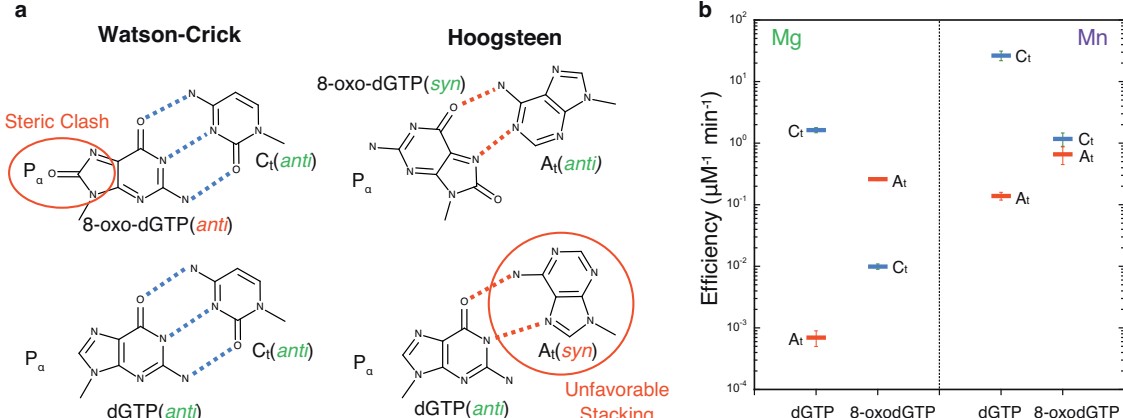

**Fig. 1 Base-pairing and specificity of 8-oxodGTP insertion. a** Base-pairing of dGTP and 8-oxodGTP in the pol μ active site. The *anti*-conformations of undamaged dGTP and template cytosine ($C_t$) enable Watson–Crick base-pairing (bottom left). The steric and electrostatic clash (red circle) between O8 and $P_\alpha$ of 8-oxodGTP(*anti*) opposite $C_t$(*anti*) discourages adoption of the *anti*-conformation (top left). Adoption of the *syn*-conformation enables favorable 8-oxodGTP Hoogsteen base-pairing with template adenine ($A_t$) in *anti*-conformation (top right). Incorporation of dGTP(*anti*) opposite $A_t$(*syn*) creates unfavorable (red circle) interactions in the template strand (bottom right). Watson–Crick hydrogen bonding is shown with blue dashed lines, while Hoogsteen interactions are shown as red dashed lines. The conformation of each base is indicated in parentheses. **b** Catalytic efficiencies of dGTP and 8-oxodGTP insertion opposite templates C ($C_t$) or A ($A_t$) in the presence of $Mg^{2+}$ (left panel) or $Mn^{2+}$ (right panel) (see Supplementary Table 1). Catalytic efficiency ($k_{cat}/K_M$, in $\mu M^{-1}$ $min^{-1}$) is indicated as a blue line for insertion opposite $C_t$ and as a red line for insertion opposite $A_t$. Error bars shown represent standard errors (S.E.) derived from triplicate measurements.

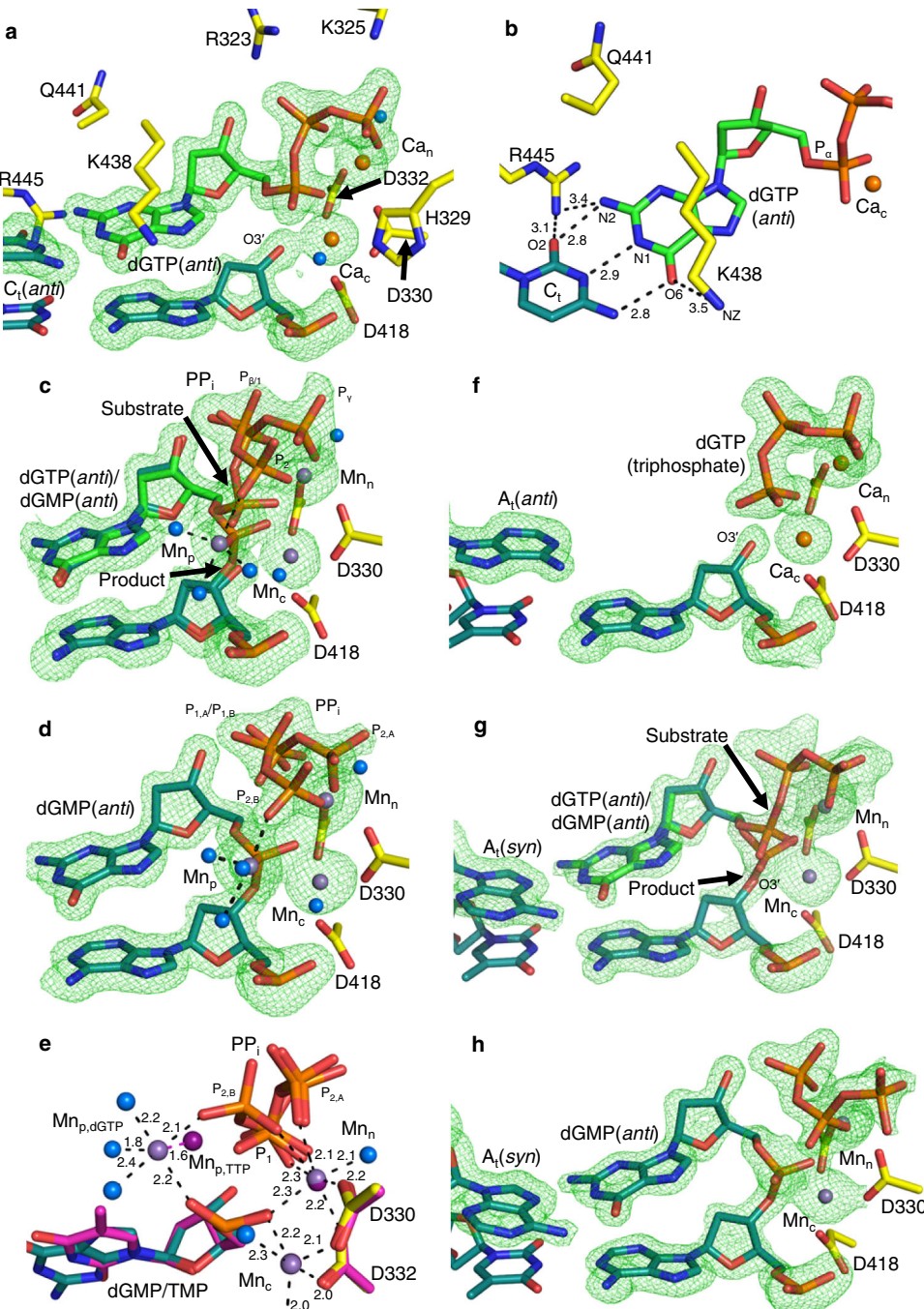

**Fig. 2 dGTP insertion opposite adenine and cytosine. a** Active site of the Ca$^{2+}$-bound dGTP(*anti*):C$_t$(*anti*) ground state ternary complex (PDB id 7KSS). Protein side chains are shown in yellow stick representation, incoming dGTP is shown in green, and DNA is in cyan. Ca$^{2+}$ atoms are orange spheres and water molecules are blue spheres. Simulated annealing omit (F$_o$-F$_c$) density (green mesh) shown here and in subsequent panels is contoured at 3.0σ, carve radius 2.0 Å. **b**, Top down close up view of dGTP(*anti*) interactions in the active site. Hydrogen bonding is shown with black dashes and distances (Å) are labeled. **c**, The Mn$^{2+}$-reaction state (RS) ternary complex of the dGTP(*anti*):C$_t$(*anti*) insertion (PDB id 7KST) after a 2 min soak in a cryo solution containing 10 mM MnCl$_2$. Arrows indicate bond broken (substrate) and formed (product). Mn$^{2+}$ atoms had exchanged for Ca$_n$ and Ca$_c$ and are shown as purple spheres. **d**, The Mn$^{2+}$-product state (PS) ternary complex of the dGTP(*anti*):C$_t$(*anti*) insertion (PDB id 7KSU). Ternary complex crystals were soaked in a cryo solution containing 10 mM MnCl$_2$ for 4 min. **e**, Active site metal coordination in matched purine and pyrimidine product state ternary complexes. Shown is the TTP(*anti*):A$_t$ Mn$^{2+}$-product complex (PDB id 5TYX, in purple) overlaid with the Mn$^{2+}$:dGTP(*anti*):C$_t$(*anti*) product ternary complex (PDB id 7KSU). The distance between product metals Mn$_{p,TTP}$ and Mn$_{p,dGTP}$ is indicated with a purple dashed line. Coordination is shown with black dashes and distances are labeled. **f**, Active site of the Ca$^{2+}$-bound dGTP:A$_t$(*anti*) ground state ternary complex (PDB id 7KSZ) after a 960 min soak in a cryo solution containing 20 mM CaCl$_2$ and 2 mM dGTP. The triphosphate is visible but density for the guanine base is absent. A$_t$ adopts the *anti*-conformation. **g**, The reaction state of the dGTP(*anti*):A$_t$(*syn*) insertion after 180 min of soak in an Mn$^{2+}$-containing cryo solution (PDB id 7KT1). Arrows indicate bond breakage (substrate) and formation (product). **h**, Product state (PS) of the dGTP(*anti*):A$_t$(*syn*) insertion after 225 min of soak in a Mn$^{2+}$-containing cryo solution (PDB id 7KT2).

metal (Ca$_c$) sites are occupied by Ca$^{2+}$ (Supplementary Fig. 2a). Metal coordination occurs through conserved active site aspartates 330, 332, and 418 (Fig. 2a and Supplementary Fig. 2a). The triphosphate of dGTP interacts with Gly320, Arg323, Lys325, and His329 (Fig. 2a).

To observe dGTP insertion in crystallo, pre-catalytic Ca$^{2+}$-bound dGTP(*anti*):C$_t$(*anti*) ternary complex crystals were soaked in cryo solutions containing 10 mM Mn$^{2+}$ or Mg$^{2+}$ to initiate the synthesis reaction in crystallo[24]. The crystals were plunged into liquid nitrogen to stop the reaction and the structures of the complexes were determined. Structures exhibiting DNA extension, termed reaction state (RS) ternary complexes, displayed the appearance of simulated annealing omit (F$_o$–F$_c$) density consistent with inversion of P$_\alpha$ stereochemical configuration upon P$_\alpha$–O3′ bond formation. Loss of density for the P$_\alpha$–P$_\beta$ bond was consistent with bond breakage. Occupancy refinement of incoming and incorporated nucleotides indicated ~50% insertion had occurred in the Mn$^{2+}$ (2 min, Fig. 2c, Supplementary Table 2) or Mg$^{2+}$ (10 min, Supplementary Fig. 2b, Supplementary Table 2) soaks. Mn$^{2+}$ (Mn$_n$) and Mg$^{2+}$ (Mg$_n$) had exchanged for Ca$_n$ in structures determined from the Mn$^{2+}$ and Mg$^{2+}$ soaks, respectively. Mn$^{2+}$ had exchanged for Ca$_c$ in the Mn$^{2+}$ soak (Mn$_C$, Fig. 2c), while based on coordination distances (1.8–2.8 Å), a mixture of Mg$^{2+}$ and Na$^+$ (Mg$_c$/Na$_c$) occupied the catalytic site in the Mg$^{2+}$ reaction state structure (Supplementary Fig. 2b). Additional density consistent with a third metal appeared in structures determined from soaks with both metals (Mn$_{p,dGTP}$, Fig. 2c; Mg$_{p,dGTP}$, Supplementary Fig. 2b). The metal coordinated phosphates of product oxygens on the inserted dGMP and pyrophosphate (PP$_i$) product, as well as three water molecules (Fig. 2c and Supplementary Fig. 2b). PP$_i$ could be modeled in a conformation rotated ~90° from that expected directly after cleavage, where P$_2$ of PP$_i$ (former Pγ of dGTP) interacted with the product metal (Fig. 2c and Supplementary Fig. 2b). Active site side-chain interactions with dGTP(*anti*) remained identical to the GS complex, apart from Asp330, that could be modeled in two conformations in the Mg$^{2+}$ (Supplementary Fig. 2b), but not the Mn$^{2+}$ structure (Fig. 2c). Asp330 coordinated both catalytic and nucleotide metals, or had rotated ~90° into a conformation exhibiting a longer Mg$_c$ coordination distance. Longer soaks in Mn$^{2+}$ (4 min, Fig. 2d, Supplementary Table 2) and Mg$^{2+}$ (30 min, Supplementary Table 2) cryo solutions generated the product state (PS) ternary complexes, that displayed active site stabilization similar to the ground and reaction state complexes. Na$_c$ was bound in the Mg$^{2+}$ soak, the product metal had departed the Mg$^{2+}$ (Supplementary Fig. 2c), but not Mn$^{2+}$ product complex (Fig. 2d and Supplementary Fig. 2d), and metal binding remained otherwise identical to the RS complexes. The electron density of Mg$^{2+}$ at partial occupancy is indistinguishable from a water molecule, so the anomalous signal emitted by Mn$^{2+}$ was used to confirm product metal binding[12,14,16,19,24,26] (Supplementary Fig. 2d). The location of Mn$_p$ was shifted ~1.6 Å compared to TTP insertion (Fig. 2e). PP$_i$ could be modeled in two distinct conformations in the Mn$^{2+}$ (Fig. 2d), but only in the rotated conformation in the Mg$^{2+}$ product complex (Supplementary Fig. 2c). Asp330 had rotated in the Mg$^{2+}$, but not Mn$^{2+}$, product complex (Fig. 2d and Supplementary Fig. 2c).

Active site interactions and positions of residues in the Ca$^{2+}$-bound pre-catalytic ground state dGTP:A$_t$ ternary complex were identical to those observed in the Ca$^{2+}$:dGTP(*anti*):C$_t$(*anti*) complex (Fig. 2f, Supplementary Table 3). A$_t$ adopted the *anti*-conformation, consistent with the binary complex[25], and although the triphosphate was fully bound, density for the guanine base was absent. The mispaired A$_t$ adopted the *syn*-conformation in the Mn$^{2+}$:dGTP(*anti*):A$_t$(*syn*) reaction (180 min, Fig. 2g) and product (225 min, Fig. 2h) ternary complexes. While bond formation and cleavage, as well as the exchange of Ca$_n$ and Ca$_c$ for Mn$^{2+}$, were clearly observed, the pyrophosphate region

was dynamic and could not be accurately modeled. Side-chain dynamics were similar to the corresponding matched insertion intermediates (Fig. 2c, d). Bond formation was not observed in the 60 min Mg$^{2+}$ soak (Supplementary Fig. 2e, Supplementary Table 3), density for the base was still absent, and A$_t$ remained in the *anti*-conformation. The structure was otherwise similar to the Ca$^{2+}$–GS complex, apart from the exchange of Ca$_n$ and Ca$_c$ for Mg$^{2+}$.

**8-oxodGTP insertion opposite adenine.** The 8-oxodG base adopts the *syn*-conformation in the pol β[13] and λ[15] active sites to allow 8-oxodGTP to evade polymerase fidelity checkpoints. To uncover molecular insights into 8-oxodGTP discrimination strategies employed by pol μ, we solved the structure of the Ca$^{2+}$-bound pre-catalytic 8-oxodGTP:A$_t$ ground state (GS) ternary complex by soaking pol μ-DNA binary complex crystals with a template A (A$_t$) in a cryo solution containing 2 mM 8-oxodGTP in the presence of 20 mM Ca$^{2+}$ for 120 min (Fig. 3a, b, Supplementary Table 3). The *syn*-conformation of the 8-oxodG base allows hydrogen bonding with A$_t$(*anti*) using its Hoogsteen edge. N2 and an oxygen of P$_\alpha$ (~2.7 Å), as well as N3 and O5′ (~3.1 Å), form stabilizing hydrogen bonds (Fig. 3b). Lys438 is within hydrogen-bonding distance of O6 (~3.3 Å), while van der Waals interactions with its methylene backbone stabilize the *syn*-conformation. O8 interacts with Arg445 through a water molecule in the minor groove. This GS structure is thus very similar to the Ca$^{2+}$–GS structure of TTP opposite A$_t$ (TTP:A$_t$, PDB id 5TXX)[24] (Supplementary Fig. 3a, b).

Ca$^{2+}$-ground state ternary complex crystals were soaked in cryo solutions containing 10 mM Mn$^{2+}$ or 50 mM Mg$^{2+}$ to initiate the reaction in the crystal. The Mn$^{2+}$-(30 min, Fig. 3c, Supplementary Table 3) and Mg$^{2+}$-(60 min, Supplementary Fig. 3c, Supplementary Table 4) 8-oxodGTP(*syn*):A$_t$(*anti*) reaction state (RS) ternary complexes displayed active site interactions largely identical to the Ca$^{2+}$–GS complex. Bond formation and breakage were evident, as well as the exchange of both Ca$_n$ and Ca$_c$ for Mn$^{2+}$ or Mg$^{2+}$, respectively. The 8-oxoG base remained in *syn*-conformation and PP$_i$ was in the position expected directly after bond cleavage, but at a reduced occupancy. In addition to the pre-catalytic conformations, Asp330 could be modeled in the rotated conformation in the Mn$^{2+}$-, but not Mg$^{2+}$-reaction state. Apart from full bond formation and breakage, the Mn$^{2+}$- (120 min, Fig. 3d, Supplementary Table 4) and Mg$^{2+}$- (180 min, Supplementary Fig. 3d, Supplementary Table 4) product state (PS) ternary complexes closely resembled the respective ground and reaction states. Density for additional metals was not observed. Asp330 was modeled as in the RS complexes. PP$_i$ was bound at reduced occupancy with Mg$^{2+}$, whereas only partial density remained with Mn$^{2+}$, precluding accurate PP$_i$ modeling.

**8-oxodGTP insertion opposite cytosine.** Hydroxyl adduction of the C8 carbon creates a steric and electrostatic clash with the sugar-phosphate of 8-oxodGTP in the *anti*-conformation. For pol β, a divalent cation is observed to alleviate this clash in the pre-catalytic ternary complex[12]. We solved the structure of the Ca$^{2+}$-ground state (GS) 8-oxodGTP:C$_t$ ternary complex after soaking pol μ–DNA binary complex crystals in the presence of 20 mM Ca$^{2+}$ and 2 mM 8-oxodGTP for 120 min. Incoming 8-oxodGTP adopts the *anti*-conformation and forms Watson–Crick hydrogen bonds with C$_t$(*anti*) (Fig. 4a, b, Supplementary Table 4). A close contact between O8, O5′ (~2.7 Å) and a non-bridging oxygen of P$_\alpha$ (~3.2 Å) is evident (Fig. 4b). Van der Waals interactions with Lys438 stabilize the *anti*-conformation, while NZ is within hydrogen-bonding distance (~3.8 Å) of O6 of the guanine base. N2 of 8-oxodG directly interacts with Arg445 in the DNA minor groove, while the latter

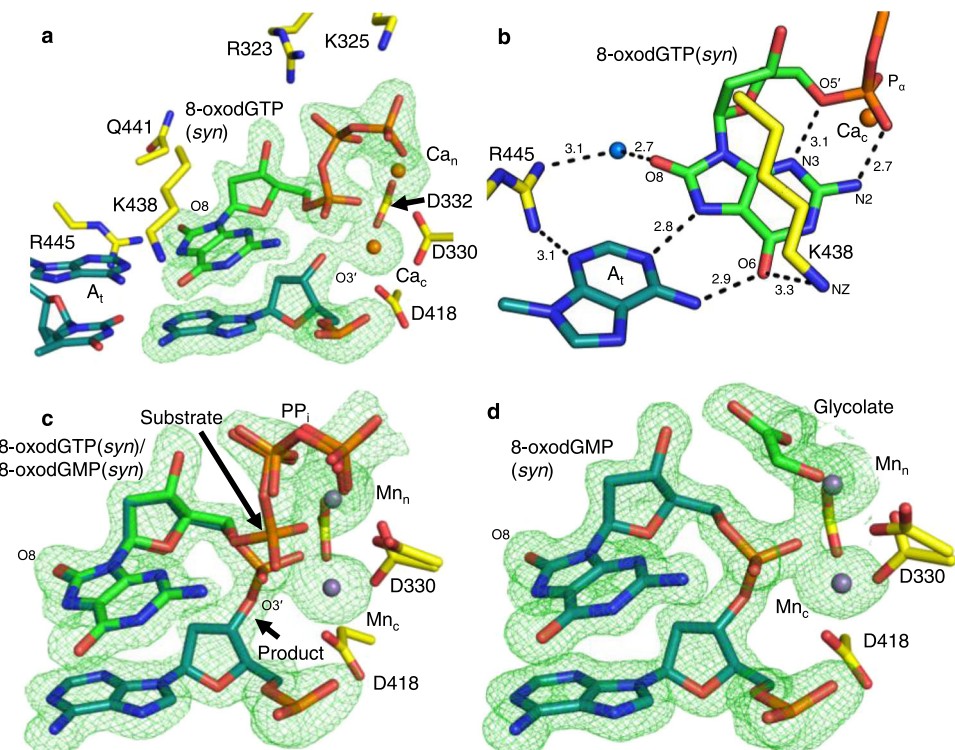

**Fig. 3 8-oxodGTP insertion opposite $A_t$. a** Active site of the 8-oxodGTP(*syn*):$A_t$(*anti*) ground state (GS) ternary complex (PDB id 7KT3). Side-chains that interact with the incoming nucleotide or metals are shown. $Ca^{2+}$ atoms are shown as orange spheres. Protein side chains are in yellow stick representation, 8-oxodGTP is shown in green, DNA in cyan. Simulated annealing omit ($F_o$–$F_c$) density is shown as a green mesh contoured at 3.0 σ, carve radius 2.0 Å. **b** Top down close-up view of 8-oxodGTP(*syn*) interactions in the active site (PDB id 7KT3). Hydrogen bonding is shown with black dashes and distances (Å) are labeled. **c** Active site of the 8-oxodGTP(*syn*):$A_t$(*anti*) reaction state (RS) ternary complex (PDB id 7KT4) after a 30 min soak in a $Mn^{2+}$-containing cryo solution. Arrows indicate bonds broken (substrate) and formed (product). $Mn^{2+}$ atoms are purple spheres. **d** Active site of the product state of the 8-oxodGTP(*syn*):$A_t$(*anti*) insertion after 120 min of soak (PDB id 7KT5). Glycolate was modeled into the density vacated by $PP_i$. Water molecules are displayed as blue spheres.

also interacts with the template base. Indeed, Arg445 is closer to N2 and Lys438 is slightly further (~0.5 Å) from O6 than opposite $A_t$ (Fig. 3b). The nucleotide and catalytic metal sites are occupied by $Ca^{2+}$, and evidence for an additional metal that might stabilize the *anti*-conformation was absent (Fig. 4a, Supplementary Fig. 4a).

Active site interactions remained consistent in the $Mn^{2+}$-(40 min, Fig. 4c, Supplementary Table 5) and $Mg^{2+}$-(90 min, Supplementary Fig. 4b, Supplementary Table 5) reaction state (RS) ternary complexes. A product manganese ($Mn_p$) was observed to coordinate product oxygens of the phosphates of 8-oxodGMP and $PP_i$ (Fig. 4c, d). Surprisingly, $Mn_p$ was shifted ~2 Å compared to TTP or dGTP insertion and now also coordinated O8 of 8-oxodGMP along with three water molecules (Fig. 4d–f). Short soaks at lower (10 mM) $MgCl_2$ concentrations lacked nucleotide incorporation. Increased (50 mM) $MgCl_2$ concentration afforded bond formation and breakage without a product metal-stabilized intermediate (Supplementary Fig. 4b). The product state (PS) ternary complexes determined from the $Mn^{2+}$-(120 min, Fig. 4g, Supplementary Table 5) or $Mg^{2+}$-(180 min, Supplementary Fig. 4c, Supplementary Table 5) soaks were identical to the reaction state complexes, except for full bond formation and breakage, as well as increased occupancy of $Mn_p$ and $PP_i$. The latter was positioned as expected directly after bond cleavage. Asp330 remained in the pre-catalytic conformation.

**Post-catalytic active site metal and product dynamics**. Longer (960 min) soaks of product complexes displayed intact correct or lesioned base pairs (Fig. 5a, Supplementary Tables 2–5). Exceptionally, $Mn_p$ and $PP_i$ were bound to the active site at ~50%

occupancy in the 960 min soak of the $Mn^{2+}$:8-oxodGTP(*anti*):$C_t$(*anti*) product complex (Supplementary Fig. 5a, b, Supplementary Table 5). Density for the product metal and $PP_i$ was lost and metal binding remained otherwise identical in all other post-catalytic complexes apart from $Mg^{2+}$:8-oxodGMP(*syn*):$A_t$(*anti*), where the nucleotide metal was lost and Asp330 was disordered. Density for $PP_i$ was lost in the 72 h post-catalytic soaks opposite either template base (Supplementary Table 6), but $Mg_n$ and $Na_c$ remained bound at reduced occupancy (Supplementary Fig. 5c, Supplementary Table 6). Additionally, product metal-mediated 8-oxodGTP(*anti*):$C_t$(*anti*) insertion was observed at sub-physiological (20 μM) concentrations of $Mn^{2+}$ in the cryo solution (Fig. 5b, Supplementary Fig. 5d, Supplementary Table 6).

**Insights into discrimination from site-directed mutagenesis**. Active site side-chains can influence nucleotide discrimination, e.g., through differential stabilization of the *syn/anti*-conformations, or by modulating active site metal dynamics. Lys438 stabilizes the guanine base through van der Waals and hydrogen bonding interactions. Lys438 was replaced with the side-chain found in a structurally equivalent position in other X-family pols (pol β, Lys438Asp; pol λ, Lys438Ala; Tdt, Lys438Arg; Supplementary Fig. 6a). Gln441 and Arg445, potential DNA minor groove hydrogen bond donors, were substituted with alanine (Gln441Ala, Arg445Ala).

The efficiency of $Mg^{2+}$-dependent dGTP:$A_t$ or :$C_t$ insertion decreased significantly compared to wild type for the Lys438Asp variant (~37- and 23-fold, respectively) (Fig. 6a, Supplementary

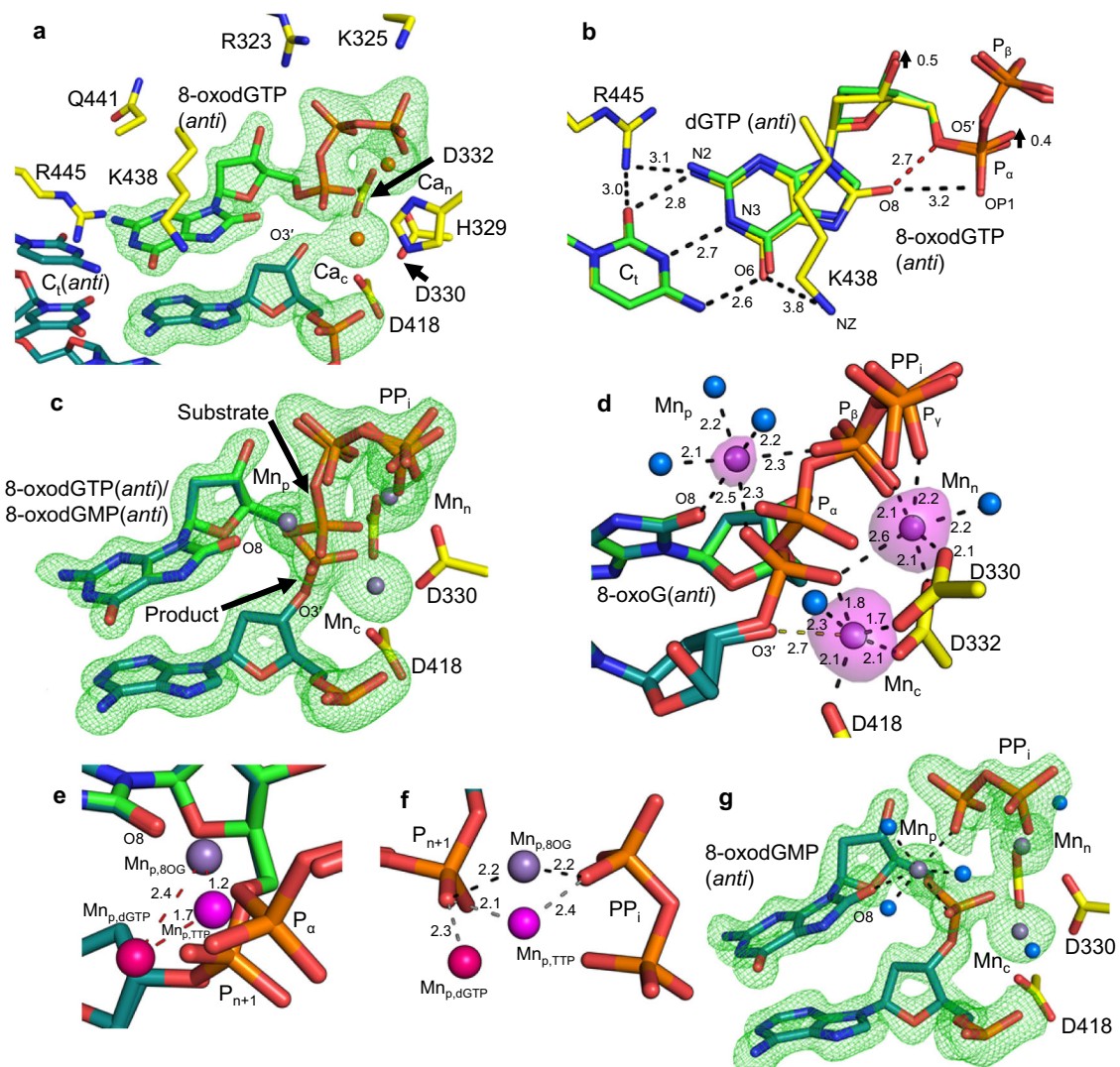

**Fig. 4 8-oxodGTP insertion opposite $C_t$. a** Active site of the $Ca^{2+}$-bound 8-oxodGTP(*anti*):$C_t$(*anti*) ground state (GS) ternary complex (PDB id 7KTA). Active site residues that interact with the incoming nucleotide or metals are shown. $Ca^{2+}$ atoms are displayed as orange spheres. Protein side chains are shown in yellow stick, DNA is in cyan, and the incoming nucleotide is in green. Simulated annealing ($F_o$–$F_c$) omit density (green mesh) shown is contoured at 3.0 σ, carve radius 2.0 Å. **b** Top down close-up view of 8-oxodGTP(*anti*) interactions in the active site (PDB id 7KTA). 8-oxodG and template bases (yellow) are shown overlaid with the $Ca^{2+}$:dGTP(*anti*):$C_t$(*anti*) ground state complex (green). Arrows indicate differences in the positions of dGTP and 8-oxodGTP. Hydrogen bonding for 8-oxodGTP is shown with black dashes and distances (Å) are labeled. Clashes are shown with red dashes. **c** Active site of the $Mn^{2+}$-reaction state (RS) ternary complex of pol μ inserting 8-oxodGTP(*anti*) opposite $C_t$ (PDB id 7KTB). $Ca^{2+}$:8-oxodGTP:$C_t$ ternary complex crystals were soaked in a $Mn^{2+}$-containing cryo solution for 40 min. Arrows indicate bond broken (substrate) and formed (product). $Mn^{2+}$ atoms are purple spheres. **d** Active site metal coordination in the RS complex (PDB id 7KTB). Coordination is shown with dashes and distances (Å) are labeled. Water molecules are shown as blue spheres. Anomalous density (magenta surface) is contoured at 5 σ. **e** Comparison of the locations of $Mn_{p,8\text{-}oxodGTP}$ (purple) (PDB id 7KTB), $Mn_{p,dGTP}$ (pink) (PDB id 7KSU), and $Mn_{p,TTP}$ (magenta) (PDB id 5TYX) displayed overlaid on the $Mn^{2+}$:8-oxodGTP(*anti*):$C_t$(*anti*) reaction state ternary complex (PDB id 7KTB). Red dashes indicate distances between metal ions. **f** Comparison of $Mn_{p,8OG}$ (purple) (PDB id 7KTB), $Mn_{p,dGTP}$ (magenta) (PDB id 7KSU), and $Mn_{p,TTP}$ (pink) (PDB id 5TYX) coordination by the incorporated phosphate ($P_{n+1}$) and phosphate oxygen of $PP_i$ in the 8-oxodGTP(*anti*):$C_t$(*anti*) reaction state ternary complex (PDB id 7KTB, 8-oxodGTP not shown for clarity). Coordination distances (Å) are shown with black dashes for $Mn_{p,8OG}$ and with grey dashes for $Mn_{p,dGTP}$ and $M_{np,TTP}$. **g** Active site of the product state of the 8-oxodGTP(*anti*):$C_t$(*anti*) insertion (PDB id 7KTC). Ground state ternary complex crystals were soaked in a $Mn^{2+}$-containing cryo solution for 120 min.

Fig. 6b, Supplementary Table 7). The efficiency and fidelity of the other variants were only modestly affected (<10-fold, Supplementary Fig. 6b). $Mn^{2+}$ generally increased dGTP insertion opposite both templates relative to $Mg^{2+}$ (Fig. 6a), whereas fidelity was decreased (Supplementary Fig. 6b). The catalytic inefficiency of dGTP insertion opposite $C_t$, observed with the Lys438Asp variant in the presence of $Mg^{2+}$, was restored to near wild-type levels in the presence of $Mn^{2+}$ (Fig. 6a), resulting in increased fidelity (Supplementary Fig. 6b). The

Arg445Ala variant displayed the lowest fidelity due to a significant increase in dGTP misinsertion in the presence of $Mn^{2+}$ (Fig. 6a). Insertion of 8-oxodGTP opposite $A_t$ was significantly increased, and that opposite $C_t$ significantly decreased, in the presence of both $Mg^{2+}$ and $Mn^{2+}$ (Fig. 6b, Supplementary Table 8). This resulted in a substantial drop in fidelity (Supplementary Fig. 6c). In contrast to the fidelity of dGTP insertion (Supplementary Fig. 6b), $Mn^{2+}$ increased the fidelity of 8-oxodGTP insertion except for the Lys438Asp variant

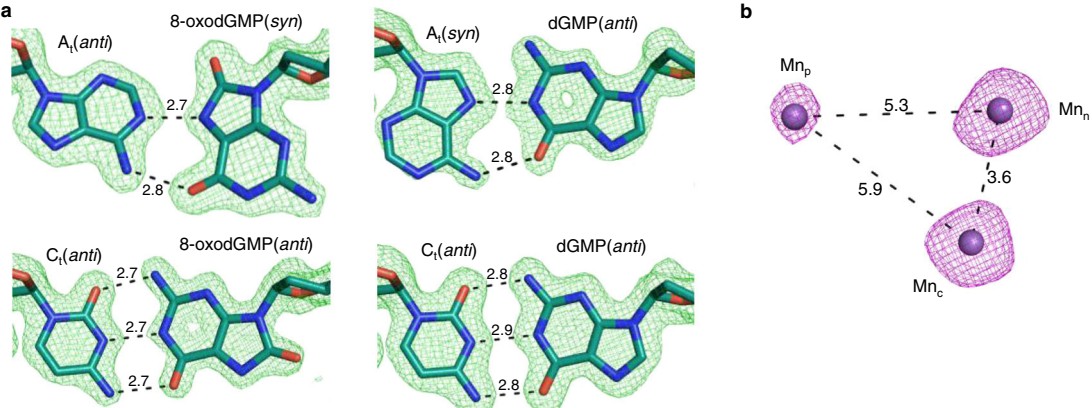

**Fig. 5 Active site dynamics. a** Base-pair hydrogen bonding is retained post-catalysis. Hydrogen bonding with the template base in the 960 min $Mn^{2+}$ soaks of the 8-oxodGMP(*syn*):$A_t$(*anti*) (PDB id 7KT6), 8-oxodGMP(*anti*):$C_t$(*anti*) (PDB id 7KTD), dGMP(*anti*):$C_t$(*anti*) (PDB id 7KSV), and the dGMP(*anti*): $A_t$(*syn*) (PDB id 7KT2) product complexes are shown with black dashes and distances (Å) labeled. DNA is shown in cyan stick representation. $F_o$-$F_c$ density (green mesh) shown is contoured at 3.0 σ, carve radius 2.0 Å. **b** Product metal supports catalysis at sub-physiological $Mn^{2+}$ concentrations (PDB id 7KTI). Ground state $Ca^{2+}$:8-oxodGTP(*anti*):$C_t$(*anti*) ternary complex crystals were soaked in a cryo solution containing 20 μM $Mn^{2+}$ for 120 min. Full product formation has occurred. Anomalous map (purple mesh, contoured at 4 σ) displays density for $Mn_p$. Distances between metal atoms are shown with dashes and distances (Å) are indicated.

relative to that observed with $Mg^{2+}$ (Supplementary Fig. 6c). However, the fidelity of 8-oxodGTP insertion was always significantly lower than that observed with dGTP.

To explain these observations, we solved the structure of the pre-catalytic $Ca^{2+}$:8-oxodGTP:$C_t$ ground state ternary complex of the Lys438Asp variant (Fig. 6c, Supplementary Table 6). Surprisingly, 8-oxodGTP adopted the *syn*-conformation and formed Hoogsteen-like hydrogen bonds with $C_t$(*anti*). The primer terminus had rotated ~180° relative to wild type (Fig. 6c, Supplementary Fig. 7b, c), while still stacking with the 8-oxodG(*syn*) base. O3′ of the rotated primer terminus was now stabilized by $Ca^{2+}$ (instead of $Na^+$) bound to the HhH motif through a water molecule (Supplementary Fig. 7b). The loss of steric restraints due to removal of the primer terminus prompted Trp434 to rotate into the major groove (Supplementary Fig. 7d). Apart from Lys438, active site interactions remained similar to wild type (Fig. 6c, Supplementary Fig. 7c). Snapshots of $Mn^{2+}$-mediated 8-oxodGTP:$C_t$ insertion displayed an active site organization (Fig. 6d, e) identical to equivalent wild-type inter-mediates (Fig. 4). The reaction state (RS, 30 min) displayed a product metal stabilized *anti*-conformation of the 8-oxoG base, enabling Watson–Crick hydrogen bonding with $C_t$(*anti*) (Fig. 6d). The coordination of the product metal was identical to wild-type, but the carboxyl side-chain of Asp438 now interacted with the product metal through a coordinating water molecule (Fig. 6e). Density for the flipped primer terminus remained in the RS, but density for the *syn*-conformation was absent. The $Mn^{2+}$-product (PS, 90 min) complex was similar to the $Mn^{2+}$-RS apart from increased occupancy of $Mn_p$ (Supplementary Table 6) and loss of density for the $P_\alpha$–$P_\beta$ bond. Subsequent to an $Mg^{2+}$ soak (90 min), $Mg_n$, and $Mg_c$ were bound but product formation was not observed and active site interactions remained similar to the $Ca^{2+}$–GS complex (Fig. 6f). $Mg^{2+}$ was thus unable to stabilize the *anti*-conformation or O3′ in the absence of Lys438, and catalysis halted in the pre-catalytic state.

## Discussion

Nucleotide salvage pathways recycle or degrade oxidized or otherwise damaged dNTPs as a result of oxidative stress-induced damage to the cellular dNTP pool[27]. The pyrophosphorylase MutT/MTH1, for example, hydrolyzes 8-oxodGTP, preventing incorporation during DNA replication and repair[6–8]. Despite a number of 8-oxodG specific repair pathways, elevated levels of

this damaged lesion promote human disease[28,29]. DNA poly-merase discrimination is crucial to prevent pro-mutagenic 8-oxodGTP insertion, especially during error-prone double-strand break repair. Employing atomic resolution time-lapse crystal-lography and kinetic analyses, we uncover the molecular attri-butes that facilitate and deter discrimination of pro-mutagenic 8-oxodGTP insertion by DSB repair polymerase μ.

The increased stability of 8-oxodG(*syn*) arises from positioning of the C8 oxygen away from deoxyribose oxygens[30]. The opposing or template base, however, appears to dictate glycosidic preference in the confines of duplex DNA or the polymerase active site (Figs. 1a, 5a). Numerous structural studies have highlighted 8-oxodG accommodation in the templating position[5,13,14,31–34]. Structural characterization of $Mg^{2+}$-medi-ated 8-oxodGTP(*syn*):$A_t$ insertion by just DNA polymerases β[12] and λ[15] have been reported, while the 8-oxodGTP(*anti*):$C_t$ insertion has only been characterized for pol β[12]. The open but rigid pol μ active site[25] appears to constrain nascent base pair geometry to modulate productive insertion. Perturbing the nas-cent base pair with damaged nucleotides or mismatches results in altered active site geometry, decreasing base pair stability and influencing chemistry.

Hoogsteen base-pairing facilitates replication of oxidized and undamaged mispairs by pol μ. Preferential stabilization of 8-oxodGTP in *syn*-conformation enables efficient 8-oxodGTP(*syn*): $A_t$(*anti*) insertion (Fig. 3)[24]. The clash of O8 with $P_\alpha$ is thus avoided, while additional stabilizing interactions are provided through stacking with the primer terminal base, as well as through interactions of N2 and N3 with $P_\alpha$ and O5′, respectively. While $A_t$ adopts the *anti*-conformation in the undamaged and oxidized dGTP:$A_t$ pre-catalytic ternary complexes, consistent with that observed in the binary complex[25], rotation of $A_t$(*anti*) into the *syn*-conformation, observed concurrently with bond formation, is required for stabilization of the dGTP(*anti*):$A_t$(*syn*) misinsertion (Fig. 2f–h and Supplementary Fig. 2e). This unique strategy satisfies the geometric requirements of the rigid pol μ active site. A similar strategy is employed by pol ι to increase efficiency of replication past templating purines[35,36]. In pol ι, however, Hoogsteen base-pairing in the TTP(*anti*):$A_t$(*syn*) insertion is observed in the pre-catalytic ternary complex. This likely enhances rather than deters productive insertion, as $A_t$(*syn*) would not be required to alter conformation during synthesis.

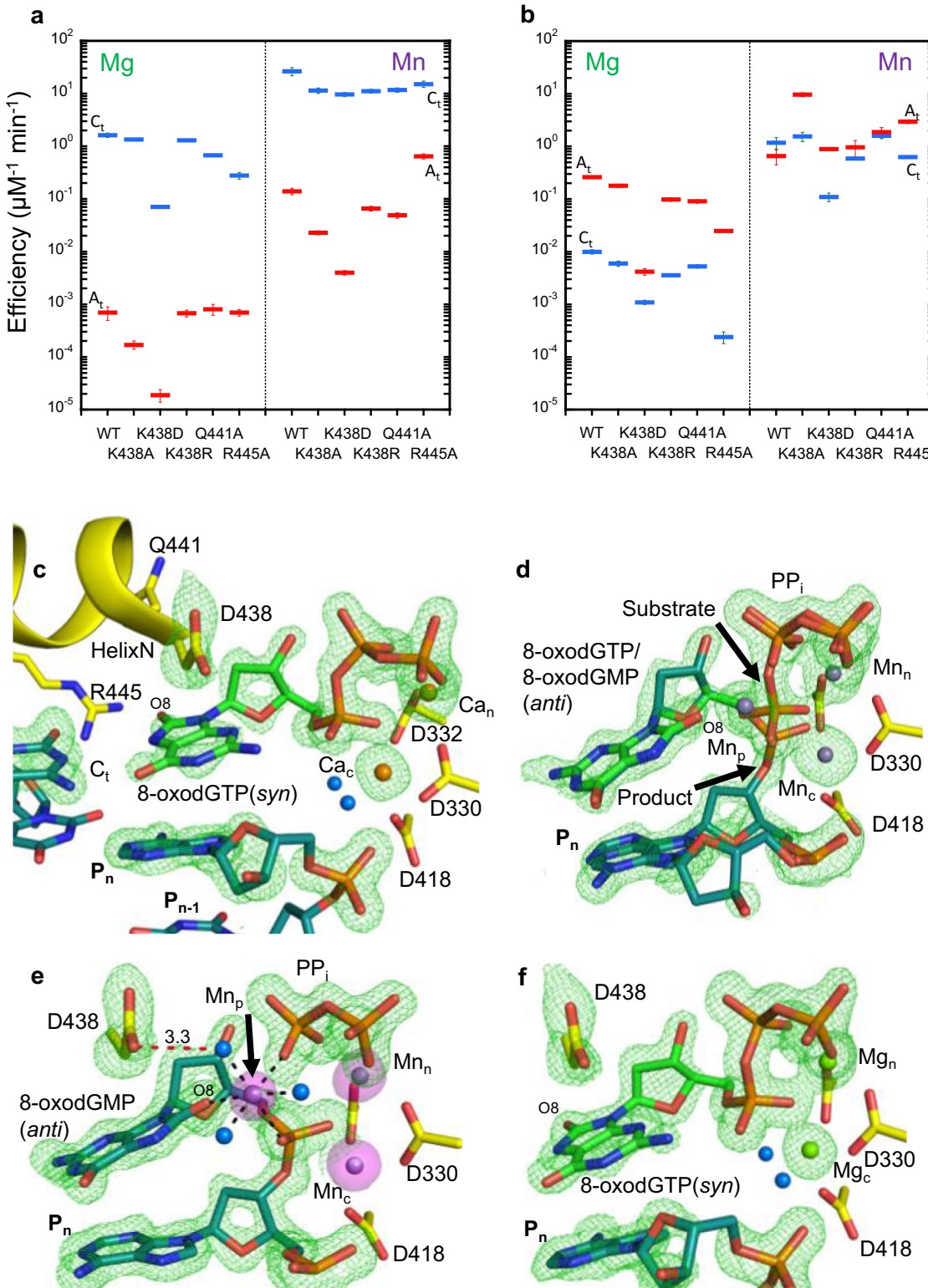

The decreased nascent base pair geometry in pol μ likely creates strain on the template strand, resulting in energetic and kinetic barriers to productive insertion. The unfavorable geometry appears to influence metal binding at the catalytic site that is overcome by $Mn^{2+}$-(Fig. 2g, h), but not $Mg^{2+}$-(Supplementary Fig. 2e) mediated stabilization of the strained conformation leading to misinsertion at reduced efficiency compared to the 8-oxodGTP(syn):$A_t$(anti) insertion (Fig. 1b). Although hydrogen-

bonding of the 8-oxodG(syn):$A_t$(anti) base pair mimics the TTP (anti):$A_t$(anti) Watson–Crick base pair[24] (Supplementary Fig. 3a, b), product metal binding is not observed with either $Mg^{2+}$ or $Mn^{2+}$ (Fig. 3c, d and Supplementary Fig. 3c, d), unlike for pol β[12]. Indeed, 8-oxodG lacks ligands able to bind $Mn_p$ in the syn-conformation and N2 partially overlaps with this site (Fig. 3b). Additionally, the favorable geometry of the 8-oxodGTP(syn): $A_t$(anti) base pair alters the geometric and charge environment of

**Fig. 6 Insights into discrimination from site-directed mutagenesis.** Catalytic efficiencies ($k_{cat}/K_M$, in $\mu M^{-1}$ $min^{-1}$) for **a** dGTP (see Supplementary Table 7) and **b** 8-oxodGTP (see Supplementary Table 8) insertion by pol μ active site variants opposite $C_t$ (blue line) or $A_t$ (red line) determined in the presence of $Mg^{2+}$ (left panel) or $Mn^{2+}$ (right panel). The error bars represent standard errors (S.E.) from triplicate measurements. **c** Active site of the $Ca^{2+}$-bound 8-oxodGTP(*syn*): $C_t$(*anti*) ground state ternary complex (PDB id 7KTJ) of the Lys438Asp variant after a 120 min soak of pol μ/Lys438Asp-DNA binary complex crystals in a cryo solution containing 2 mM 8-oxodGTP and 20 mM $CaCl_2$. The Lys438Asp active site of the 8-oxodGTP(*anti*): $C_t$(*anti*) insertion after a 120 min soak in 20 mM $CaCl_2$/2 mM 8-oxodGTP and transfer to a cryo solution containing 50 mM $MnCl_2$ for **d** 30 min (reaction state, PDB id 7KTM), and **e** 90 min (product state, PDB id 7KTL). Hydrogen bonding between Asp438 and a water molecule coordinating $Mn_p$ is shown with a red dashed line. **f** Active site of the $Mg^{2+}$-bound 8-oxodGTP (*syn*): $C_t$(*anti*) ternary complex of the Lys438Asp variant (PDB id 7KTK) after soaking as in **e** and **f** but instead of the $MnCl_2$ soak, ground state $Ca^{2+}$-bound ternary complex crystals were soaked in a cryo solution containing 50 mM $MgCl_2$ for 90 min. In the above panels (**c–f**), simulated annealing $F_o$–$F_c$ density (green mesh) shown is contoured at a contour level of 3.0 σ, carve radius 2.0 Å, while anomalous density (magenta surface) is shown at 5 σ.

$PP_i$ compared to undamaged insertion, facilitating $PP_i$ dissociation from the active site (Fig. 3d).

Although 8-oxodGTP is accommodated in the active site in the *syn*-conformation opposite $C_t$ in the absence of Lys438 (Fig. 6a and Supplementary Fig. 7), efficient 8-oxodGTP:$C_t$ insertion appears to require 8-oxodG to base-pair using the Watson–Crick edge (Fig. 6d–f). 8-oxodGTP thus adopts the *anti*-conformation, as in the undamaged dGTP(*anti*):$C_t$(*anti*) insertion (Fig. 2). This is energetically unfavorable due to electrostatic and steric clashes with $P_α$ (Fig. 4b). Whereas pol β mediated insertion of 8-oxodGTP (*anti*) requires an additional ground state metal to neutralize the clash with $P_α$[12], pol μ can bind 8-oxodGTP(*anti*) in its absence (Fig. 4a, b). The oxidized base is positioned closer to the templating cytosine and away from the sugar–phosphate backbone compared to dGTP(*anti*):$C_t$(*anti*) in the rigid pol μ active site, resulting in shorter Watson–Crick hydrogen bonds (Fig. 5a), compared to other G:C duplex base pairs, or e.g., in pol β. Additionally, the nascent base pair buckles more severely than in pol β, reducing electrostatic conflicts with the sugar–phosphate backbone (Fig. 4b). Watson–Crick incorporation opposite $C_t$ is thus preferred by the undamaged guanine in the absence of this strain, but deterred by more favorable stabilization of the 8-oxodG (*anti*) base in the rotated orientation of the primer terminus, resulting in decreased stability of O3′, likely due to decreased catalytic metal binding (Supplementary Fig. 7b–d). Reduced catalytic metal binding decreases the stability of O3′ and hinders proton abstraction, thus reducing incorporation. Primer terminal instability appears to be hindered by Lys438, as the primer terminus is more readily displaced in the ground state 8-oxodGTP (*syn*):$C_t$(*anti*) ternary complex of the Lys438Asp variant (Fig. 6c and Supplementary Fig. 7b, d). The appearance of $Mn_p$ in the 8-oxodGMP(*anti*):$C_t$(*anti*) product complex, bridging O8 with two product oxygens and three water molecules, promotes insertion by stabilizing the poor geometry of the *anti*-conformation (Fig. 4b). The metal was observed opposite both templates in pol β, but not opposite an undamaged mispair. This effect can be partially overcome by $Mn_p$ stabilization of the *anti*-conformation, and thus the primer terminus, when accompanied by $Mn_c$ but not $Mg_c$ binding, even at low concentrations (20 μM, Fig. 5b and Supplementary Fig. 5d). Higher $Mg^{2+}$ concentrations (50 mM) enable sufficient O3′ stabilization without a product metal-stabilized intermediate, likely due to increased $Mg_c$ binding (Supplementary Fig. 4b, c). This provides sufficient stabilization of the *anti*-conformation to enable insertion in the absence of $Mg_p$. Since $Mg^{2+}$ does not promote efficient 8-oxodGTP(*anti*):$C_t$(*anti*) insertion (Fig. 1b), these observations suggest that base-pair geometry in the rigid pol μ active site is strained in the *anti*-conformation and a product metal is required to stabilize insertion. Better substrate geometry in the 8-oxodGTP(*syn*):$A_t$(*anti*) insertion promotes productive alignment of catalytic moieties even in the absence of $Mn_p$. Decreased discrimination of 8-oxodGTP insertion thus relies on a mechanism that takes advantage of the rigid pol μ active site architecture and metal dynamics.

Increased overall levels of 8-oxodG insertion by pol μ likely promotes pro-mutagenic DSB repair. Additional pathways, such as OGG1- and MutY-mediated repair are presumably recruited to repair the incorporated lesions, as replication of unrepaired ligated lesions would lead to transversion mutations[11,29]. The absence of an abnormal base pair following 8-oxodGTP insertion opposite either template base (Fig. 5a) would enhance downstream DSB processing, such as ligation. In contrast, 8-oxodGTP insertion by pol β[12] creates abortive ligation intermediates that require further processing[37]. Pol μ, therefore, mediates stabilization of cytotoxic DSBs to promote NHEJ repair.

Side-chain interactions with the base of the incoming nucleotide influence the efficiency and discrimination of nucleotide insertion by pol μ. The guanine base is sandwiched between Lys438 and the base of the primer terminus. $Mg^{2+}$-dependent dGTP insertion is impacted by the Lys438Asp variant and $Mn^{2+}$ rescued insertion opposite $C_t$ but not $A_t$ (Supplementary Table 7). Discrimination was therefore the same in the presence of either metal, whereas other variants exhibited lower fidelity in the presence of $Mn^{2+}$, as is generally observed for DNA polymerases. In contrast to dGTP, the identity of the metal had the opposite effect on 8-oxodGTP discrimination. $Mg^{2+}$- rather than $Mn^{2+}$-dependent insertion exhibited lower fidelity, however, the specificity of the Lys438Asp mutant was similar with both metal cofactors. In either case, both metals exhibited very low fidelity with 8-oxodGTP. Mutagenesis of residues equivalent to Lys438 in pol λ (Asn510) and pol β (Asp276) has been shown to influence 8-oxodGTP[15] and Fapy-dGTP[38] discrimination, respectively. Lys438 provides steric constraints for the incoming nucleotide and primer terminus, as substitution of this residue with aspartate results in 8-oxodG(*syn*) and primer terminal displacement (Supplementary Fig. 7). The effect can be rescued by $Mn^{2+}$, but not $Mg^{2+}$ (Fig. 6, Supplementary Table 8). Asp438 surprisingly interacts with $Mn_p$ through a water molecule (Fig. 6e), likely providing increased stabilization and thus efficiency of $Mn^{2+}$-dependent insertion. This is the first observation of the interaction of a protein side-chain with the product metal. For wild-type, the improved $Mn^{2+}$-dependent specificity is primarily due to an increase in the catalytic efficiency for insertion opposite $C_t$, rather than a decrease in insertion opposite $A_t$, as the insertion efficiency of 8-oxodGTP opposite $A_t$ was similar with both metals (Fig. 1b, Supplementary Table 8). Lys438 thus has a significant role in nucleotide selectivity in the pol μ active site. The role of Trp434 in active site stabilization is additionally revealed in the Lys438Asp mutant, where this residue contributed to modulation of catalysis by destabilizing the primer terminus (Supplementary Fig. 7).

Arg445 interacts directly with N2 of 8-oxodGTP(*anti*) opposite $C_t$, and through a water molecule with O8 opposite $A_t$, promoting Watson–Crick and Hoogsteen base pairing with the template base, respectively. The interaction of Arg445 stabilizes 8-oxodGTP incorporation, and ensures that a base pair with the proper geometry is formed in the active site. The equivalent residues in pols β (Arg283)[13] and λ (Arg517)[15] appear to play similar roles. The limited decrease in fidelity with the Arg445Ala

mutant contrasts with the stronger effect of the equivalent substitution in pol β[12,39]. This may be in part due to the lack of subdomain repositioning in pol µ upon nucleotide binding[25]. Pol λ also exhibits less pronounced subdomain motions during ternary complex formation, and nucleotide binding is accommodated by shifts in the template strand[40]. Arg517 appears to play key roles in template strand alignment, and the Arg517Ala substitution decreases the frameshift fidelity of pol λ[41].

Due to its conserved nature in the pol X active site, Gln441 and equivalent residues in pols β (Asn279) and λ (Asn513)[42] (Supplementary Fig. 6a), is expected to strongly influence nucleotide stabilization in the active site. Gln441 does not directly interact with the guanine base in the pol µ active site, but interacts with O8 of 8-oxoG(syn) in pols β[12] and λ[15], as well as with N3 in the anti-conformation. Gln441Ala substitutions alter 8-oxodGTP discrimination in pols β[43] and λ[15], but not in pol µ (Fig. 6b, Supplementary Table 8). The limited fidelity effect observed here suggests a decreased contribution to the stabilization of nucleotide insertion.

The DNA polymerase reaction is reversible such that the forward (synthesis) reaction is in chemical equilibrium with the reverse reaction[44]. The reverse (chemical) reaction generates a primer shortened by one dNMP ($DNA_{-1}$) and an intact triphosphate. Indirect estimates of the chemical equilibrium are lower than for the overall reaction[45,46]. The polymerase thus perturbs the equilibrium to facilitate the forward (chemical) reaction. Yang et al. described a third active site metal in Y-family polymerase η[19,26]. This third transient metal coordinates product phosphate oxygens of the incorporated dNMP and $PP_i$, and was suggested to be essential for DNA synthesis by lowering the energy barrier for the synthesis reaction[19]. This product metal was also observed in post-catalytic complexes of X-family pols β[16] and µ[24]. Unlike with pol β, the presence of $Mn_p$ during matched (dGTP and TTP[24]) insertion in pol µ correlates with increased insertion efficiency compared to the $Mg^{2+}$-mediated reaction. In the latter, synthesis still occurs, but $Mg_p$ is absent. Similarly, observation of $Mn_p$ (Fig. 4), but not $Mg_p$ (Supplementary Fig. 4b, c), correlates with increased efficiency of 8-oxodGTP(anti):$C_t$ insertion (Fig. 1b), where its altered location (Fig. 4e, f) likely stabilizes the primer terminus (Fig. 6d, e and Supplementary Fig. 7), as well as the product complex, retaining $PP_i$ in the active site (Supplementary Fig. 5a). Efficient synthesis was observed in the presence of higher (50 mM) $Mg^{2+}$ concentrations, where increased binding to the catalytic site likely stabilizes the primer terminus (Supplementary Fig. 4b, c). The product metal was observed in the 8-oxodGTP(syn):$A_t$ insertion in pol β[12], but not µ (Fig. 3). The product metal, therefore, inhibits the reverse reaction, promoting the forward (synthesis) reaction, consistent with stable product complexes (DNA/$PP_i$) of pols µ or β that do not form dNTP and $DNA_{-1}$, and computational studies[17,18,47]. Notably, a $PP_i$ analog (PNP) that decreases the chemical and overall equilibrium, permits crystallographic characterization of the reverse reaction[48]. Additionally, a stable metal-bound pol β/dNTP/DNA complex can be formed in the absence of the product metal.

Subsequent to nucleotidyl transfer, the catalytic metal dissociates in the $Mg^{2+}$-, but not $Mn^{2+}$-mediated, undamaged reactions due to loss of an $Mg_c$ coordinating ligand[24] (Fig. 2d and Supplementary Fig. 2b, c). Required to align catalytic atoms to enable the reverse reaction[18], the loss of $Mg_c$, and exchange for $Na_c$ blocks the reverse reaction, shifting the overall reaction equilibrium towards synthesis. Similarly, as $Mn_c$ remains bound, $Mn_p$ binding is required to stabilize the product complex and prevent the reverse reaction, as $PP_i$ is retained in the active site (Supplementary Fig. 5a, b). $Mg_c$ appears longer lived in the 8-oxodGTP insertions (Supplementary Figs. 3c, 3d, 4b, 4c); $PP_i$ is lost immediately upon formation opposite $A_t$ due to altered active site geometry (Fig. 3d) and is unavailable to

undergo the reverse reaction, removing the need for the product metal. $Mn_c$ remains bound in the 8-oxodGTP(syn):$A_t$(anti) reaction and product ternary complexes despite the loss of a ligand due to rotation of Asp330 (Fig. 3c, d), as $Mn^{2+}$ can accommodate variable coordination spheres more readily than $Mg^{2+}$ [49,50]. Density for $Mg_p$ is absent in the 8-oxodGTP(anti):$C_t$ insertion even at the higher (50 mM) concentration (Supplementary Fig. 4b, c), while $Mg_c$ remains bound, resulting in low efficiency for this insertion (Fig. 1b, Supplementary Table 1). Manganese has indeed been suggested to be the in vivo metal cofactor of pols µ[21,22] and λ[51], and is activated in vitro at physiological $Mn^{2+}$ concentrations (i.e., 50 µM)[21,52–57], but inhibited at physiological $Mg^{2+}$ concentrations (1–2 mM)[58]. We observed full product metal-mediated synthesis at sub-physiological $Mn^{2+}$ concentrations (20 µM) (Fig. 5b), suggesting manganese[21–23], and $Mn_p$, may promote the biological role of pol µ in DSB repair.

## Methods

**Protein expression and purification.** Truncated human pol µ (loop 2 deletion mutant, hPol µ Δ2) was overexpressed in BL21-CodonPlus(DE3)-RIL cells (Invitrogen) at 16 °C overnight and purified as previously described[25]. Cells were lysed by sonication in 25 mM Tris/HCl, pH 8.0 (25 °C), 500 mM NaCl, 5% glycerol, 1 mM DTT. Pol µ was batch purified on Glutathione Sepharose CL-4B resin (GE Healthcare) and eluted by TEV cleavage overnight at 4 °C. Size-exclusion chromatography was performed on a Superdex 200 26/600 column in 25 mM Tris/HCl, pH 8.0 (25 °C), 100 mM NaCl, 5% glycerol, 1 mM DTT, 1 mM EDTA. Pol µ was then dialyzed into 25 mM Tris/HCl, pH 8.0 (25 °C), 100 mM NaCl, 5% glycerol, 1 mM DTT, concentrated to 11 mg ml[−1], and stored at −80 °C. Expression plasmids were sequenced in both directions to confirm the expected sequence by Genewiz, Inc.

**DNA preparation.** A 9-mer template oligonucleotide (5′-CGGCXTACG-3′, where X = A or C) was mixed with 4-mer upstream (5′-CGTA-3′) and 5′-phosphorylated downstream 4-mer (5′-pGCCG-3′) oligonucleotides in a 1:1:1 ratio in 100 mM Tris/HCl, pH 7.5 (25 °C) to create the duplex DNA for crystallization[24]. The mixed oligonucleotides were heated to 95 °C for 5 mins and then cooled down to 4 °C at a rate of 1 °C/min and kept on ice until use.

A 34-mer template oligonucleotide (3′-GACGTCGACTACGCGXCATGCCTA GGGGCCCCATG-5′, where X = A or C) was annealed with a 17-mer upstream and a 15-mer 5′-phosphorylated downstream oligonucleotide complementary to the template sequence to create the duplex DNA for kinetic assays. The annealing buffer consisted of 10 mM Tris/HCl, pH 7.5 (25 °C), 1 mM EDTA. Oligonucleotides were mixed in a 1.2:1.2:1 ratio (template:downstream:upstream) and annealed as above. PAGE purified DNA was from Integrated DNA Technologies.

**Time-lapse crystallography.** Binary complex crystals of pol µ with template adenine or cytosine in a 1-nucleotide gapped DNA were grown by mixing pol µ (11 mg ml[−1]) with mother liquor (100 mM HEPES/NaOH, pH 7.5 (25 °C), 16–18% PEG 4000) at 4 °C using the sitting-drop vapor-diffusion method[25]. Unless otherwise noted, time-lapse crystallography was performed by soaking pol µ-DNA binary complex crystals in a cryo solution containing $CaCl_2$ and incoming nucleotide (15% ethylene glycol, 100 mM HEPES/NaOH, pH 7.5 (25 °C), 20% PEG4000, 5% glycerol, 50 mM NaCl, 1–2 mM dNTP, and 10 mM $CaCl_2$). Unless otherwise noted, ground state ternary complex crystals were then soaked for increasing times in a cryo solution containing 10–50 mM $MgCl_2$ or 10 mM $MnCl_2$ (15% ethylene glycol, 100 mM HEPES/NaOH, pH 7.5 (25 °C), 20% PEG4000, 5% glycerol, 50 mM NaCl), preceded by a pre-soak wash. The crystal was then plunged into liquid nitrogen.

As revealed by the appearance of simulated annealing ($F_o$–$F_c$) omit density and occupancy refinement, full occupancy of dGTP opposite $C_t$ was achieved after a 20 min soak[25] in a cryo solution containing $Ca^{2+}$, while 8-oxodGTP required longer soak times (120 min) and higher dNTP concentrations (2 mM). The $Ca^{2+}$-bound pre-catalytic dGTP:$A_t$ ternary complex required longer (960 min) soak times likely due to lower dGTP binding affinity for the pol µ:$A_t$ binary complex active site. We thus soaked binary complex crystals directly in the secondary $Mg^{2+}$ or $Mn^{2+}$ soak that included 1 mM incoming 8-oxodGTP to obtain the reaction and product state ternary complexes.

**Data collection and refinement.** Data collection was performed at the Advanced Photon Source (Argonne National Laboratory, Chicago, IL) on the BM22 or ID22 beamlines (Southeast Regional Collaborative Access Team, SER-CAT) using the Mar225, Mar300HX, or Eiger 16M[59] detectors at a wavelength of 1.00 Å. Data were processed and scaled using the programs HKL2000[60] or HKL3000[61]. Initial models were determined using molecular replacement with a previous structure of pol µ (PDB id 4M04[25]). Refinement and iterative model building were done using the

Phenix software package[62] and Coot[63]. $R_{free}$ flags were taken from the starting model and occupancy refinement was performed, including grouped occupancy refinement of active site moieties. Simulated annealing omit ($F_o$–$F_c$) density maps were generated after partial deletion of the model in the region of interest and, unless otherwise noted, are contoured at 3 σ, carve radius 2.0 Å. Metal atoms were modeled based on electron density, coordination geometry, and coordination distances[49,50,64]. Coordination distances for $Mg^{2+}$ (1.8–2.1 Å) and $Mn^{2+}$ (2.1–2.3 Å) are typically shorter than for $Na^+$ (2.4–2.7 Å). Additionally, bound $Mn^{2+}$ was confirmed through the presence of anomalous density. Unless otherwise noted, anomalous maps are contoured at 5 σ, carve radius 2.0 Å. Ramachandran analysis determined 100% of non-glycine residues lie in allowed regions and at least 96% in favored regions. Figures were prepared in PyMol (Schrodinger) or ChemDraw (PerkinElmer).

**Gap-filling kinetic assays**. Kinetic assays were performed to measure the apparent rate ($k_{cat,app}$) and equilibrium Michaelis ($K_{m,app}$) constants of insertion. Assays were performed in a buffer containing 50 mM Tris/HCl, pH 7.4 (25 °C), 100 mM KCl, 10% glycerol, 100 µg ml$^{-1}$ bovine serum albumin, 1 mM dithiothreitol, 0.1 mM EDTA, and 10 mM $MgCl_2$ or 1 mM $MnCl_2$. Concentrations of $Mg^{2+}$ or $Mn^{2+}$ were adjusted to account for metal binding by dNTP. Pol µ was pre-incubated with 100 nM single-nucleotide gapped DNA and mixed with dNTP to initiate the reaction. Reactions were quenched with 0.25 M EDTA and mixed with an equal volume of formamide dye. Reaction products were then separated on denaturing (18%) gels and quantified using a Typhoon imager and Fluoroimager 595.

**Reporting summary**. Further information on research design is available in the Nature Research Reporting Summary linked to this article.

## Data availability

Atomic coordinates and structure factors for the reported crystal structures have been deposited in the Protein Data Bank (PDB) under accession numbers: 7KSS, 7KST, 7KSU, 7KSV, 7KSW, 7KSX, 7KSY, 7KSZ, 7KT0, 7KT1, 7KT2, 7KT3, 7KT4, 7KT5, 7KT6, 7KT7, 7KT8, 7KT9, 7KTA, 7KTB, 7KTC, 7KTD, 7KTE, 7KTF, 7KTG, 7KTH, 7KTI, 7KTJ, 7KTK, 7KTL, 7KTM, and 7KTN. All data is available from the authors upon reasonable request.

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

## Acknowledgements

The authors wish to thank the South-East Regional Collaborative Access Team (SER-CAT) for assistance with data collection and the NIEHS Mass Spectrometry and Research Support Group for help with MS/MS analysis. We thank Drs. Lars Pedersen, Kasia Bebenek, Pavel Afonine, Kiyoe Ura, and Mr. Hiroaki Tanuma for helpful suggestions and/or technical assistance. Use of the Advanced Photon Source (APS) was supported by the U.S. Department of Energy, Office of Science, Office of Basic Energy Sciences, under contract W-31-109-Eng-38. This research was supported by research project numbers Z01-ES050158, Z01-ES050161 (S.H.W.), and 1K99ES029572-01 (J.A.J) in the intramural research program of the National Institutes of Health-NIEHS, and JSPS KAKENHI Grant Number 16K16195 (A.S.).

## Author contributions

J.A.J. performed the crystallography. A.S. and D.D.S. performed the kinetic analysis. J.A.J., A.S., D.D.S., W.A.B., and S.H.W. performed data analysis. J.A.J., W.A.B., and S.H. W. wrote the paper. S.H.W. supervised the research.

## Competing interests

The authors declare no competing interests.
