## [Peer Review File · Nature Communications]

Reviewers' comments:

Reviewer #1 (Remarks to the Author):

This paper offers intriguing evidence that mutagenic mismatch pairing of 8oxoG with A by pol mu requires a third metal. This third metal is coordinated to the DNA itself and was discovered only through time-resolved crystallography. This technique has revealed these type of metals in polymerases and nucleases in recent years. In previous cases though, the third metal has been involved in stabilizing the "proper" reaction. As far as I can discern, this paper represents the first case of an "improper" pairing promoted by the third metal and thus provides a substantive new consideration in the polymerase field. The authors have done a substantial amount of crystallographic work with an impressive 32 data sets with such high resolution and provided a detailed analysis.

The one major concern is whether this third metal can be considered physiological (discussed in more detail below), and this concern is based on the ambiguity of the metal concentrations used in this study. For the field, the authors should address this issue and should consider adding in additional biochemistry experiments that examines the mispairing propensity as a function of metal concentrations.

- a) Electron density maps in all figures should be simulated annealing omit maps, to provide experimental support of the model, and defined as such in figure legend and main text. If carving applied, this should be noted.
- b) It was difficult to figure out the Mg and Mn concentrations used in each of the crystallization and biochemistry experiments in the methods, crystal tables, and the figure legends, with the exception of Figures 2 and 5. In one part of the paper, 10 mM Mg²⁺ was defined as low (with 10 mM Ca²⁺ present?). Please include metal concentrations for every experiment, including the concentrations in the soaks for each crystal table.
- c) How is the misincorporation of adenine affected by Mg/Mn concentration in vitro? One can argue that the high concentrations are needed for rapid and complete occupation of the metal site during the soak in the mother liquor of the crystal (I'm guessing high concentration from what I could discern in main text). Presumably, biochemical experiments would not require such high concentrations and provide support for the physiological relevance of the result. I noted that the final paragraph states the physiological concentrations but was unable to locate it in the referenced paper. I recommend finding the original paper that measures the metal concentration in the cells. For Figure 5, what is the calculated occupancy, if the B factors are defined as the same as the high concentration soaks?
- d) Is the third site a higher affinity site than the protein-coordinated two Mg sites or vice versa...
- e) Are there mixed occupancies in the time-resolved? A table documenting change in relative population would be helpful to readers. If possible, incorporating % into figures would also aid understanding
- f) Without stereo of the metal sites, it's difficult to ascertain coordination angles. Please do stereo in the extended figures 4 and 5
- g) Please include WT in extended data tables 3 and 4. I
- h) Given the number of structures in this manuscript, it would be helpful to readers to have the pdb associated with a particular figure in the figure. That way if they want to look up the structure themselves, it's much easier.
- i) While Mg and Mn are often considered interchangeable, albeit with different catalytic rate impacts, it is interesting that Mn promotes a completely different base pair geometry than Mg. As the structures are such high resolution, can the authors explain at the level of metal coordination geometry the discrepancy between Mg and Mn in promoting mispairing? Why isn't the mispairing occurring in Mg²⁺?
- j) The subscript next to the metals was obscure as to how they were defined (c for catalytic?, n for native?). Please define these in the text.
- k) How was the Na and Mg atoms identified in the crystal structure? Mn has an anomalous signal, which makes it easier to detect than Mg. Please include in methods
- l) It is tempting to think there might be an advantage of pairing 8oxoG against adenine? It would seem advantageous to simply prevent this mispairing otherwise. Do the authors have any ideas? Perhaps where oxidation is used beneficially. One possibility might be in antibody differentiation? Pol mu has been linked to light chain differentiation. Is there a different mn concentration in

- iimmune cells? Is there a link between Pol mu and antibody appearance in evolution.
- m) Is there anything that might indicate specificity for a deoxy vs a ribonucleotide for the 8oxoG insertion?
- n) Undamaged insertion as the figure title is a bit obscure. Watson:Crick or G:C insertion may be more accessible.
- o) Are the text in the figures minimally ten point? Some are difficult to read.
- p) It is unusual to use mass spec to validate mutations. Was the entire gene sequenced as well?
- q) What was the annealing buffer for the 13+4 mer DNA substrate.
- r) Why is the concentration for crystallization 1-3 mM dNTP? Was it different for each NTP? It would be better to provide exact concentration or have it documented on the table.

Reviewer #2 (Remarks to the Author):

This manuscript explores the mechanism by which pol mu discriminates against 8-oxodGTP using time-lapse x-ray crystallography. The data appear to be of high quality and the interpretation of the results is sound. The manuscript reads very well, although it is understandably a bit dense considering there are 32 different x-ray structures that are being described. Overall, the manuscript does an excellent job at highlighting how nucleotide discrimination occurs through changes in the positioning of nucleotide/template, and the presence or absence of either Mg²⁺ or Mn²⁺, and the relative position of water molecules-- all while the protein itself is held rigid.

-It may be beneficial to the reader to add an explanation pertaining to the replacement of Mg²⁺ with Mn²⁺ in the crystallographic and fidelity measurements earlier in the manuscript (in results section, rather than the end of the discussion). Readers in the field will definitely grasp the significance right away, but others may question the purpose of carrying out experiments with Mn²⁺.

-Most of the results are discussed in relation to the "rigid active site" of pol mu. In the first instance of this, reference 24 is given, but it would be better if a sentence or two were used to explain what exactly is meant by this.

-The Km for correct nucleotide incorporation (dGTP across from C) in the presence of Mn²⁺ is very low compared to Mg²⁺ (6 nM versus 3.5 uM, respectively). For the DNA polymerases that I'm familiar with there isn't nearly this large of a difference with Mg/Mn. Has this been observed before with this polymerase, or other low-fidelity polymerases?

-It would be nice if Figure 6A and B could be widened so that the labels aren't overlapping. Or maybe angle the labels so they all fit on the level?

-including an amino acid sequence alignment for pol mu, beta, lambda, and Tdt in the supplemental might be helpful for the 'substitution analysis' section.

-The discussion of how metals dictate the equilibrium of the reaction is really interesting. A discussion of the kinetics of the process might help though. Does increasing the Mg²⁺ concentration increase the reverse reaction?

Reviewer #3 (Remarks to the Author):

This manuscript by Jamsen and colleagues presents a thorough characterization of the catalytic cycle of incorporation of oxidized and undamaged nucleotides by DNA polymerase mu. Taking advantage of time-resolved x-ray crystallography and a robust system for carrying out catalysis in crystallo, the authors solve numerous crystal structures that explain how DNA polymerase mu facilitates incorporation of dGMP or 8oxodGMP opposite template A or C. The structures help identify the roles of specific side chains, the differences in catalysis depending on whether

manganese or magnesium are used as cofactors, as well as the importance of a third catalytic metal to determine the fidelity of 8oxodGMP incorporation. The structural observations are nicely supported by kinetic measurements of nucleotide incorporation. The manuscript is well-written, has a thoughtful discussion and provides substantial insight into the mechanism of catalysis of DNA polymerase μ and other family X polymerases. The presentation of the data is clear and the figures are well-made. I only have a few minor suggestions that might help improve the manuscript.

The statistical analyses presented are adequate and a sufficiently thorough description of the methods is provided.

Minor points

-A comment in the introduction highlighting the relevance of studying the incorporation of oxidized nucleotides by DNA polymerase μ would be helpful for readers

-Similarly, although this is discussed in the last sentences of the discussion, it would be helpful for readers not in the field to comment on the relevance of magnesium vs manganese for catalysis by DNA polymerase μ in the introduction (or perhaps in the results section).

-I find it confusing that the main text does not mention that ternary complexes are obtained through soaking of binary complexes (this is explained in the methods but also in the figure legends). I would suggest clarifying this in the text. Also, since presumably the authors have solved the corresponding binary structures it might be useful to mention if, as expected, the conformation of the templating A or C is consistent with that observed in the pre-catalytic complexes (i.e., anti).

-The titles of the subheading seem unnecessarily terse. I would suggest revising them to make them more informative (for instance, "insertion opposite adenine" could refer to the undamaged insertion).

We thank the reviewers for their careful reading, constructive remarks, and insightful comments. A detailed response to each comment is given below. Changes in the manuscript file are highlighted in yellow. We believe the revised manuscript has been improved and is suitable for publication.

REVIEWER COMMENTS

REVIEWER #1

Major comment: The one major concern is whether this third metal can be considered physiological (discussed in more detail below), and this concern is based on the ambiguity of the metal concentrations used in this study. For the field, the authors should address this issue and should consider adding in additional biochemistry experiments that examines the mispairing propensity as a function of metal concentrations.

Response: The function and physiological relevance of the third or product metal is unknown. The metal is observed for correct insertion with X-family pols μ (Ref. #24) and β (Refs. #16), as well as a member of the Y-family, pol η (Ref. #19, 26). It has not been observed for an incorrect insertion. The product metal was suggested to lower the transition state barrier for the forward DNA synthesis reaction (Refs. #19, 26), or deter the reverse (pyrophosphorolysis) reaction (Refs. #12, 16, 17, 18, 24). Both suggestions essentially alter the equilibrium of the chemical step towards product. Since 8-oxodGTP in *syn*-conformation mimics TTP(*anti*), it was of interest whether a third metal would promote the forward reaction. One goal of this study was thus to characterize the structural attributes that modulate product metal binding in response to damaged substrates.

The concentration of metals in the crystallographic soaks and kinetic experiments were given under Methods. Mn^{2+} has been suggested to modulate the biological function of pol μ (Refs. #21, 22), but the physiological metal is unclear. Mn^{2+} can be easily discerned from its anomalous signal crystallographically compared to Mg^{2+} or Na^+ . Accordingly, we examined whether active site metal binding sites were occupied at low Mn^{2+} concentrations. We found that not only were all three sites occupied but that full product formation had taken place. These observations suggested that even very low, sub-physiological Mn^{2+} concentrations enable nucleotide insertion. The concentration of the metals in the crystallographic soaks and kinetic experiments are further clarified in the Results of the revised manuscript, in the Methods, in the figure legends, in each PDB file, and in the crystallographic tables (Supplementary Tables 2-6).

The physiological relevance of the product metal would depend on the specific role of this metal, the physiological concentrations of metals present, and their specific binding affinities. The third metal is a transient product-associated metal and thus associates with, and dissociates from, the product complex. The affinity of the site therefore depends on the extent of the reaction and the state of dissociation of PP_i . Qualitative estimates with pol η suggest that Mg^{2+} binding to the 3rd site is significant (Ref. #19) under cellular concentrations of this metal. Equilibrium binding affinities of each site have yet to be quantitatively determined, however, and a specific kinetic effect has so far not been definitively demonstrated. Experiments with variable metal concentrations would also be challenging to interpret since the identity, affinity, or occupancy of the relevant metal binding sites would be unknown. Kinetic studies as a function of metal ion and nucleotide concentration complemented with mutagenesis (pol μ binds up to 8 Mn in our structures) to confirm the identity of the binding site(s), would significantly complicate the analysis.

The manuscript already includes extensive characterization of 8-oxodGTP insertion opposite templates C (C_i) and A (A_i) with wild-type and 5 mutant enzymes in the presence of either Mg^{2+} or Mn^{2+} . Further characterization would make the manuscript untenable for the average reader.

Minor comment (a): Electron density maps in all figures should be simulated annealing omit maps, to provide experimental support of the model, and defined as such in figure legend and main text. If carving applied, this should be noted.

Response: Unless specifically noted, the green mesh in all figures represents simulated annealing F_o-F_c electron density maps contoured at 3σ . Anomalous maps are presented as a magenta surface contoured at 5σ . Density was carved at 2.0 \AA to enhance clarity. Increased carve radii did not alter maps in the regions of interest. These details are noted in the Methods section of the revised manuscript (p. 28, lines 577-583), and further clarified in the Figure legends.

Minor comment (b): It was difficult to figure out the Mg and Mn concentrations used in each of the crystallization and biochemistry experiments in the methods, crystal tables, and the figure legends, with the exception of Figures 2 and 5. In one part of the paper, 10 mM Mg^{2+} was defined as low (with 10 mM Ca^{2+} present?). Please include metal concentrations for every experiment, including the concentrations in the soaks for each crystal table.

Response: See Response to Major comment above. The Ca^{2+} , Mg^{2+} and Mn^{2+} soak concentrations are stated explicitly in the Methods section (p. 26-27, lines 547-564) of the manuscript and clarified at appropriate locations in the Results, Discussion and Figure legends (see e.g. p. 5 lines 85-88, p. 6 lines 99-100, p. 9 lines 164-166, p. 9 lines 175-176, p. 10 lines 198-200, p.11 lines 217-223, p. 46 lines 974-981 and p.46 lines 984-986) of the revised manuscript, in each PDB file, and in the crystallographic data tables (Supplementary Tables 2-6). The time-lapse protocol is described in the Results (e.g. p. 5, lines 84-88, p.6 98-100) of the revised manuscript. For further details, please refer to our previous publication (Ref. #24) and other sources (Refs. #12, 14, 16, 19, 26).

Minor comment (c): Response is divided to cover several sub-comments

Minor comment (c1): How is the misincorporation of adenine affected by Mg/Mn concentration in vitro? One can argue that the high concentrations are needed for rapid and complete occupation of the metal site during the soak in the mother liquor of the crystal (I'm guessing high concentration from what I could discern in main text). Presumably, biochemical experiments would not require such high concentrations and provide support for the physiological relevance of the result.

Response: Incorporation opposite adenine appears to be less affected in the crystal than opposite C_t . This correlates with kinetic results, where Mn^{2+} increases the catalytic efficiency of insertion opposite C_t to a greater extent than opposite A_t (Fig. 1b). As indicated in Methods (p. 28 lines 589-594), the solution studies were performed at concentrations of Mg^{2+} (10 mM) and Mn^{2+} (1 mM) typically employed by laboratories in the field and provide for direct comparison with other studies. Defining the metal concentration dependence of fidelity would require altering the concentration of both the metal as well as the incoming nucleotide. While these experiments are feasible, they are outside the scope of this study (see response to Major comment above). The manuscript already includes extensive kinetic characterization with wild-type and mutant enzymes.

Minor comment (c2): I noted that the final paragraph states the physiological concentrations but was unable to locate it in the referenced paper. I recommend finding the original paper that measures the metal concentration in the cells.

Response: The physiological Mn^{2+} concentration (20-53 μM) is given in references 56-57 of the revised manuscript, whereas Mn^{2+} concentration quoted in references 52-55 of the revised manuscript is the total level that elicits toxicity in dopaminergic neurons (60-150 μM).

Minor comment (c3): For Figure 5, what is the calculated occupancy, if the B factors are defined as the same as the high concentration soaks?

Response: The B-factors in the higher (10 mM) and lower (20 μM) Mn^{2+} concentration soaks are approximately 2-fold different. We altered the B-factor of Mn_p in the 20 μM soak to the higher value and refined atomic positions and occupancies. As the result, we obtained an occupancy value of 0.9. Negative density surrounded Mn_p , indicating that the occupancy at the higher B-factor was over-estimated and not

consistent with experimental density. The high-resolution permits modeling of occupancies and B-factors in the structures reported in Supplementary Tables 2-6. Additionally, the B-factors for the metal ions are consistent with occupancies and B-factors of surrounding atoms. As outlined in the Results and Discussion of the manuscript, the claim regarding Figure 5b relates strictly to the observation of product formation and of a product metal (Mn_p) at low, sub-physiological, Mn^{2+} concentrations. Support for these observations is presented in the form of strong (F_o-F_c , Supplementary Fig. 5d) density demonstrating bond formation and the presence of active site metal atoms (F_o-F_c and anomalous density, Fig. 5b, Supplementary Fig. 5d).

Minor comment (d): Is the third site a higher affinity site than the protein-coordinated two Mg sites or vice versa...

Response: A decrease in anomalous density of Mn_p during the *in crystallo* metal titration experiment, but less of a decrease at the catalytic and nucleotide metal sites, suggests this site has lower affinity than the nucleotide and catalytic metal sites (Fig. 5b, Supplementary Tables 2 & 5). This is supported by the occupancies and B-factors of metal sites in the Mg soaks. However, as we have shown here, the affinity of the third metal site may depend on the insertion under study and other factors. Gao and Yang (Ref. #19) estimated the third metal binding site displays the lowest affinity for natural matched nucleotide insertion. However, since product metal binding requires both products (inserted dNMP and PP_i), binding is only observed in the product complex, and the metal dissociates upon longer soaks. The affinity of the site therefore depends on the extent of the reaction, and the state of dissociation of PP_i . Additionally, a kinetic approach to determining the affinities of each site is outside the scope of this study (see response to Major comment).

Minor comment (e): Are there mixed occupancies in the time-resolved? A table documenting change in relative population would be helpful to readers. If possible, incorporating % into figures would also aid understanding

Response: Considering that Mg^{2+} and Na^+ are difficult to distinguish based on electron density alone, we decided to model the metal in the catalytic site as a single metal to improve clarity, with the understanding that coordination distances suggest these sites contain both Mg_c and Na_c . We changed the wording to account for this discrepancy (see p. 6, lines 111-114), but where coordination distances indicate a mixture of metals, we kept the Mg^{2+}/Na^+ labeling in the Figures (Supplementary Fig. 2b).

Minor comment (f): Without stereo of the metal sites, it's difficult to ascertain coordination angles. Please do stereo in the extended figures 4 and 5

Response: Stereo views are now included in Supplementary Figures S4 and S5.

Minor comment (g): Please include WT in extended data tables 3 and 4.

Response: WT results are now included in Supplementary Tables S3 and S4 (Supplementary Tables 7 and 8 of the revised manuscript).

Minor comment (h): Given the number of structures in this manuscript, it would be helpful to readers to have the pdb associated with a particular figure in the figure. That way if they want to look up the structure themselves, it's much easier.

Response: PDB ids have been added to the figure legends of the revised manuscript.

Minor comment (i): While Mg and Mn are often considered interchangeable, albeit with different catalytic rate impacts, it is interesting that Mn promotes a completely different base pair geometry than Mg. As the structures are such high resolution, can the authors explain at the level of metal coordination geometry the discrepancy between Mg and Mn in promoting mispairing? Why isn't the mispairing occurring in Mg^{2+} ?

Response: Overall, our results suggest that Mn²⁺ optimizes base pairing and nucleotide insertion. Since nucleotide discrimination is coupled to the rate of insertion, discrimination is expected to be sensitive to metal identity. This is demonstrated most easily in the characterization of the K438D variant, where Ca²⁺ does not promote the *anti*-conformation in the K438D ground state 8-oxodGTP:C_t ternary complex, rather 8-oxodGTP adopts the *syn*-conformation and the primer terminus has changed conformation preventing insertion. To our knowledge, this is the first observation of the *syn*-conformation of 8-oxodGTP opposite C_t in the polymerase active site. Even in the *higher* (50 mM) concentration of Mg²⁺, although Mg_c is fully bound, it is not able to “capture” and stabilize O3′ to align catalytic groups for bond formation in the absence of K438 and 8-oxodGTP remains in *syn*-conformation. Mn²⁺-stimulated 8-oxodGTP(*anti*):C_t insertion with Mn_p bound, however, was almost as efficient as wild type. Mn_c therefore provides more effective stabilization than that afforded by Mg_c. Mn_p binding additionally stabilizes the product complex thus promoting product formation, and adoption of the *anti*-conformation. “Rigid” active site features and template strand positioning together with metal dynamics thus influence base pair geometry and nucleotide insertion. The above details are described in the Discussion section of the revised manuscript.

Minor comment (j): The subscript next to the metals was obscure as to how they were defined (c for catalytic?, n for native?). Please define these in the text.

Response: These are defined at the beginning of the Results (p. 5, line 93) as ‘n’ for nucleotide metal and ‘c’ for catalytic metal. These subscripts were also used in previous work (e.g. Refs. #12, 16, 24). Due to the compactness of the text further redundancy was removed, however, we included clarification in various sections of the revised manuscript.

Minor comment (k): How was the Na and Mg atoms identified in the crystal structure? Mn has an anomalous signal, which makes it easier to detect than Mg. Please include in methods

Response: These were identified according to electron density, coordination distances and coordination geometry, where on average Mg²⁺ displays octahedral coordination geometry with coordination distances of 1.80-2.20 Å (Refs. #49, 50). Na⁺ binding sites are distinct in that coordination distances are longer (2.4-2.7 Å) and typically accommodates 4-5 ligands (Ref. #64). These details have been added to Methods (p. 27, lines 580-582).

Minor comment (l): It is tempting to think there might be an advantage of pairing 8oxoG against adenine? It would seem advantageous to simply prevent this mispairing otherwise. Do the authors have any ideas? Perhaps where oxidation is used beneficially. One possibility might be in antibody differentiation? Pol mu has been linked to light chain differentiation. Is there a different mn concentration in immune cells? Is there a link between Pol mu and antibody appearance in evolution.

Response: Increased 8-oxodGTP insertion would result in increased mutagenic break synthesis. As significant cellular energy is expended in DSB repair and the cell has elaborate defense mechanisms to remove this lesion, increased insertion could result in increased overall DSB repair. Under conditions of increased ROS production, such as during cell killing by macrophages and other immune processes, or during exposure to environmental agents that induce damage, increased mutagenic repair could improve antibody repertoire and increase the probability of generating antibodies against varied antigens. However, detailed understanding of these processes is limited and more studies would be required to substantiate any direct effects.

Minor comment (m): Is there anything that might indicate specificity for a deoxy vs a ribonucleotide for the 8oxoG insertion?

Response: Since pol μ efficiently inserts ribonucleotides, 8-oxoGTP insertion is of biological interest. Accordingly, we are currently examining this insertion. Since the conformation of the ribose sugar is constrained by O2', discrimination would likely be altered compared to 8-oxodGTP.

Minor comment (n): Undamaged insertion as the figure title is a bit obscure. Watson:Crick or G:C insertion may be more accessible.

Response: As also requested by Reviewer #3 (see Minor point (5)), the Figure titles and subheadings in the Results section have been changed in the revised manuscript to address both reviewers' concerns.

Minor comment (o): Are the text in the figures minimally ten point? Some are difficult to read.

Response: Nature Communications guide to authors states that: "Figures are best prepared at the size you would expect them to appear in print. At this size, the optimum font size is between 5pt and 8pt." We re-formatted the figures and text in regards to font, font size, paragraphs, subheadings, and other details in order to enhance clarity within the Nature Communications guidelines.

Minor comment (p): It is unusual to use mass spec to validate mutations. Was the entire gene sequenced as well?

Response: The expression plasmids employed for WT and mutant recombinant protein expression in *E. coli* were sequenced in both directions to confirm the presence of the expected sequence. This has been added to Methods (p. 25, lines 523-524).

Minor comment (q): What was the annealing buffer for the 13+4 mer DNA substrate.

Response: The buffer was 100 mM Tris/HCl pH 7.5. These details are noted in the Methods section (p. 25-26, lines 527-532) of the revised manuscript and were used in previously publications (Ref. #24, 25).

Minor comment (r): Why is the concentration for crystallization 1-3 mM dNTP? Was it different for each NTP? It would be better to provide exact concentration or have it documented on the table.

Response: The soak details have been clarified in Methods (p. 26-27, lines 547-564), as well as in the Results and Figure legends of the revised manuscript.

REVIEWER #2

Minor comment (1): It may be beneficial to the reader to add an explanation pertaining to the replacement of Mg²⁺ with Mn²⁺ in the crystallographic and fidelity measurements earlier in the manuscript (in results section, rather than the end of the discussion). Readers in the field will definitely grasp the significance right away, but others may question the purpose of carrying out experiments with Mn²⁺.

Response: As the reviewer points out, Mn²⁺ was used as it emits a specific anomalous signal. Previous work suggests pol μ is a Mn-dependent enzyme (Refs. #21, 22). A comment on this is now included on p. 4 (lines 67-70) of the revised manuscript.

Minor comment (2): Most of the results are discussed in relation to the "rigid active site" of pol μ . In the first instance of this, reference 24 is given, but it would be better if a sentence or two were used to explain what exactly is meant by this.

Response: We employ the term "rigid active site" to indicate the constrained positions of the protein/DNA backbone and active site features in the ternary complex, which appear to force the incoming and template

nucleotides, as well as other active site moieties into similar positions, e.g. a consequence of which are the short hydrogen bonds in the 8-oxodGTP(*anti*):C_i insertion. In contrast, other X-family polymerases such as pols β and λ , undergo protein subdomain or DNA repositioning, respectively, in transitioning from a binary DNA complex to the ternary complex. Pol μ , however, does not undergo such repositioning and thus has a preformed “rigid active site”. We refer to previous publications where this topic is covered in more detail (Refs. #24, 25); noted on p. 16 (lines 320-324), p.17 (lines 335-336), p. 18 (lines 363-367), p. 19 (lines 390-392), p.19 (lines 394-396), p. 22 (lines 460-463) of the revised manuscript.

Minor comment (3): The K_m for correct nucleotide incorporation (dGTP across from C) in the presence of Mn^{2+} is very low compared to Mg^{2+} (6 nM versus 3.5 μ M, respectively). For the DNA polymerases that I’m familiar with there isn’t nearly this large of a difference with Mg/Mn. Has this been observed before with this polymerase, or other low-fidelity polymerases?

Response: The decrease in K_m in the presence of Mn^{2+} was observed previously for pol β (Batra VK, *et al.*, 2008); Mn^{2+} decreased the K_d for the incoming correct nucleotide ~10-fold in a single-turnover experiment (E>DNA). In a steady-state assay, the K_m reflects not only the binding affinity for the incoming nucleotide, but also the processivity of the enzyme at a specific site. The processivity is the ratio of the DNA dissociation rate constant (i.e., $k_{off,DNA}$) and intrinsic insertion rate constant (k_{pol}). Since $K_m = K_d(k_{pol}/k_{off})$, then the large decrease in K_m would indicate an increase in the rate of nucleotide insertion, a decrease in the rate of DNA dissociation and/or a decrease in the $K_{d,dNTP}$ (Beard *et al.* (1994) *J Biol Chem.* 269(45):28091-28097).

Minor comment (4): It would be nice if Figure 6A and B could be widened so that the labels aren’t overlapping. Or maybe angle the labels so they all fit on the level?

Response: The errors have been corrected.

Minor comment (5): including an amino acid sequence alignment for pol mu, beta, lambda, and Tdt in the supplemental might be helpful for the ‘substitution analysis’ section.

Response: We have added this alignment with the mutated residues indicated (Supplementary Fig. 6a).

Minor comment (6): The discussion of how metals dictate the equilibrium of the reaction is really interesting. A discussion of the kinetics of the process might help though. Does increasing the Mg^{2+} concentration increase the reverse reaction?

Response: The reversal of the chemical step (pyrophosphorolysis) with pol μ is currently being investigated. Initial results suggest that the rate of the reverse reaction is dependent on the source of PP_i (i.e., intrinsically generated from chemistry or the reaction supplemented with added PP_i , or PP_i analogs). The metal dependence is also being investigated.

REVIEWER #3

Minor point (1): A comment in the introduction highlighting the relevance of studying the incorporation of oxidized nucleotides by DNA polymerase mu would be helpful for readers

Response: A comment on this is included on p. 4 (lines 55-57) of the revised manuscript.

Minor point (2): Similarly, although this is discussed in the last sentences of the discussion, it would be helpful for readers not in the field to comment on the relevance of magnesium vs manganese for catalysis by DNA polymerase mu in the introduction (or perhaps in the results section).

Response: This is now mentioned on p. 4 (lines 67-70) of the revised manuscript (See Reviewer #2, Minor comment (1)).

Minor point (3): I find it confusing that the main text does not mention that ternary complexes are obtained through soaking of binary complexes (this is explained in the methods but also in the figure legends). I would suggest clarifying this in the text.

Response: The protocol for generating the ternary complex is given on p. 5 (lines 84-88) of the revised manuscript and this is further mentioned in the Results (see e.g. p. 6, lines 98-100; p. 9, lines 164-166; p. 9, lines 175-176; p. 10, lines 198-200) and Methods (p. 26-27, lines 547-564).

Minor point (4): Also, since presumably the authors have solved the corresponding binary structures it might be useful to mention if, as expected, the conformation of the templating A or C is consistent with that observed in the pre-catalytic complexes (i.e., *anti*).

Response: We are not aware of any binary structures of X-family polymerases with templating A or C in *syn*-conformation. A binary complex structure of pol μ in complex with template A has been published (PDB id 4M04, Ref. #25). This structure displays template A in *anti*-conformation in a “rigid” active site with limited differences to the ternary complex with bound incoming TTP. Additionally, Supplementary Fig. 2d shows A_i in *anti*-conformation opposite bound dGTP(triphosphate) after a 60 min soak in a Mg²⁺-containing cryo-solution, where density for the base is absent (p. 8, lines 155-157). We also solved binary complex structures with template A and C with alternative metals. In these structures, templates A and C are in the *anti*-conformation. As these unpublished structures are part of an ongoing project, we simply note that a previous structure of the binary complex with templating A is in the *anti*-conformation in the revised manuscript (p. 16, lines 330-334). We further indicated conformations of template bases, where relevant.

Minor point (5): The titles of the subheading seem unnecessarily terse. I would suggest revising them to make them more informative (for instance, “insertion opposite adenine” could refer to the undamaged insertion).

Response: The subheadings in the results section have been altered in the revised manuscript to provide a clearer description of their content. According to Nature Communications editorial guidelines, subheadings were removed from the Discussion (see Reviewer #1, Minor comment (n)).

REVIEWERS' COMMENTS:

Reviewer #2 (Remarks to the Author):

The authors have adequately addressed my comments. I support the publication of the manuscript without additional modifications.

Reviewer #3 (Remarks to the Author):

The authors have thoroughly addressed all my concerns and the manuscript is now suitable for publication

Point-by-Point Responses to referees comments

The authors wish to thank the reviewers for their time, effort and invaluable comments.

REVIEWERS' COMMENTS:

Reviewer #2 (Remarks to the Author):

The authors have adequately addressed my comments. I support the publication of the manuscript without additional modifications.

Reviewer #3 (Remarks to the Author):

The authors have thoroughly addressed all my concerns and the manuscript is now suitable for publication